# Fair Decisions from Calibrated Scores:
# Achieving Optimal Classification While Satisfying Sufficiency

**Etam Benger** [1]   **Katrina Ligett** [1 2]

## Abstract

Binary classification based on predicted probabilities (scores) is a fundamental task in supervised machine learning. While thresholding scores is Bayes-optimal in the unconstrained setting, using a single threshold generally violates statistical group fairness constraints. Under independence (statistical parity) and separation (equalized odds), such thresholding suffices when the scores already satisfy the corresponding criterion. However, this does not extend to sufficiency: even perfectly group-calibrated scores—including true class probabilities—violate predictive parity after thresholding. In this work, we present an exact solution for optimal binary (randomized) classification under sufficiency, assuming finite sets of group-calibrated scores. We provide a geometric characterization of the feasible pairs of positive predictive value (PPV) and false omission rate (FOR) achievable by such classifiers, and use it to derive a simple post-processing algorithm that attains the optimal classifier using only group-calibrated scores and group membership. Finally, since sufficiency and separation are generally incompatible, we identify the classifier that minimizes deviation from separation subject to sufficiency, and show that it can also be obtained by our algorithm, often achieving performance comparable to the optimum.

## 1. Introduction and Related Work

Making decisions based on predicted probabilities is a central task in machine learning, statistics, and data-driven decision-making. Given a score that estimates the probability of a positive outcome, the Bayes-optimal decision rule in the unconstrained setting is simple: apply a deterministic threshold to the score (Wasserman, 2004). This principle underlies many everyday decisions, from acting on a weather forecast to standard classification pipelines.

In high-stakes applications, however, decision making is often subject to fairness constraints. Algorithms increasingly affect outcomes in domains such as college admissions, lending, hiring, healthcare, and criminal justice, where concerns about discrimination and disparate treatment across protected groups are paramount. As a result, a substantial literature has developed around formal notions of fairness in classification, typically expressed as statistical constraints relating a prediction or decision $R$, a target variable $Y$, and a protected attribute $A$.

Three prominent criteria of algorithmic fairness are independence, separation, and sufficiency (Barocas et al., 2023). Independence (often called demographic parity) requires the decision $R$ to be statistically independent of the protected attribute $A$. Separation (including equalized odds and equal opportunity (Hardt et al., 2016)) requires equality of error rates across groups. Sufficiency (also known as predictive parity or group-calibration) requires that, conditional on the decision outcome, the prevalence of the positive label be the same across groups. These terms capture different fairness notions and are known to be mutually incompatible in general, except in trivial cases (Chouldechova, 2017; Kleinberg et al., 2017).

A common feature of these criteria is that, when applied to classification, they typically cannot be satisfied by a single, group-agnostic threshold on the score (Hardt et al., 2016; Corbett-Davies et al., 2017). Instead, satisfying independence or separation usually requires group-specific decision rules or explicit post-processing that depends on the protected attribute. An important exception arises when the score itself already satisfies the desired criterion: if a score satisfies independence or separation, then any post-processing preserves the property, and a single threshold can be used fairly.

This convenient property does not extend to sufficiency.

[1]School of Computer Science and Engineering, The Hebrew University of Jerusalem, Israel [2]The Federmann Center for the Study of Rationality, The Hebrew University of Jerusalem, Israel. Correspondence to: Etam Benger <etam.benger@mail.huji.ac.il>, Katrina Ligett <katrina.ligett@mail.huji.ac.il>.

*Proceedings of the 43rd International Conference on Machine Learning*, Seoul, South Korea. PMLR 306, 2026. Copyright 2026 by the author(s).

Even when a score is perfectly calibrated within each group—including the ideal case where the score equals the true conditional probability $P(Y = 1|X, A)$—applying a single threshold generally produces decisions that violate sufficiency (Chouldechova, 2017; Canetti et al., 2019). This phenomenon has been highlighted in both theoretical and empirical studies and plays a central role in well-known case studies such as the COMPAS recidivism risk score (Angwin et al., 2016), which takes values between 1 and 10 and is used to make binary decisions such as bail decisions.

At the same time, sufficiency has received significant attention in the algorithmic fairness literature (Barocas et al., 2023). It is an appealing notion for its compatibility with accuracy (in contrast, in general, achieving independence or separation requires sacrificing accuracy (Dwork et al., 2012)) and for its relationship to the notion of calibration.

Two main approaches have been proposed to address the problem of sufficiency in decision-making. Canetti et al. (2019) introduce classifiers that satisfy sufficiency by allowing an algorithm to abstain from making a decision on some instances. Baumann et al. (2022) show that sufficiency can be achieved without abstentions by suitable post-processing, but their analysis relies on strong assumptions about the score distribution, notably continuity and full support. In contrast, many practical systems operate with discrete, finite-valued scores, for which these assumptions do not hold. Other related works do not guarantee exact sufficiency, instead relaxing the problem either through approximate objectives (e.g., Celis et al., 2019; Delaney et al., 2024), or by enforcing only the condition for positive decisions (e.g., Zeng et al., 2022).

**Our contributions.** In this work, we study binary classification under sufficiency in the practically relevant setting of finite-valued, group-calibrated scores. Our focus is on post-processing methods of classification that take as input only a score and group membership and output a (possibly randomized) binary decision. We give an exact characterization of the set of achievable classifiers under sufficiency and use it to derive optimal fair decision rules.

We provide a geometric characterization of the feasible pairs of positive predictive value (PPV) and false omission rate (FOR) attainable by binary classifiers based on a calibrated score. This characterization yields a complete description of the "feasible region" and its boundary. Next, we characterize the intersection of the group-wise feasible regions induced by sufficiency and show how to trace its boundary efficiently. This leads to a simple post-processing algorithm[1] that constructs the optimal sufficient classifier for a wide class of objectives, including loss minimization, using only

---

[1]Code will be made available at
https://github.com/etambenger/
fair-decisions-from-calibrated-scores.

group-calibrated scores and group membership. Finally, recognizing that sufficiency and separation are generally incompatible, we study classifiers that satisfy sufficiency while minimizing deviation from separation. We show that this objective can be optimized within the same geometric framework and often yields classifiers with performance close to that of the loss-optimal solution.

## 2. Preliminaries

**Definition 2.1** (Sufficiency). Let $Y$ be a target variable and $A$ a protected attribute. A predictor $S$ of $Y$ satisfies *sufficiency* if $Y$ and $A$ are conditionally independent given $S$, that is,

$$P(Y \mid S, A) = P(Y \mid S).$$

Intuitively, sufficiency requires that the predictive meaning of a score be the same across groups. This notion is closely related to calibration.

**Definition 2.2** (Calibration and group-calibration). Let $Y$ be a binary target variable. A score $S \in [0, 1]$ is *calibrated* with respect to $Y$ if

$$P(Y = 1 \mid S = s) = s$$

for all $s \in \operatorname{supp} S$. If $A$ denotes population groups, $S$ is *group-calibrated* if

$$P(Y = 1 \mid S = s, A = a) = s$$

for all $a \in \operatorname{supp} A$, $s \in \operatorname{supp} S$ with $P(S = s, A = a) > 0$.

If $S$ is group-calibrated, then it satisfies sufficiency, and the converse also holds up to a relabeling of $S$ (see, e.g., Barocas et al., 2023). For nonconstant binary predictors, sufficiency is equivalent to predictive parity, defined next.

**Definition 2.3** (Predictive parity). Let $Y$ be binary and $R$ a nonconstant binary predictor. Define the positive predictive value and false omission rate as

$$\mathrm{PPV}(R) := P(Y = 1 \mid R = 1),$$
$$\mathrm{FOR}(R) := P(Y = 1 \mid R = 0).$$

$R$ satisfies *predictive parity*[2] if, for all $a \in \operatorname{supp} A$,

$$P(Y = 1 \mid R = 1, A = a) = \mathrm{PPV}(R),$$
$$P(Y = 1 \mid R = 0, A = a) = \mathrm{FOR}(R).$$

## 3. Feasibility in the Unconstrained Setting

We begin by characterizing all attainable $(\mathrm{PPV}, \mathrm{FOR})$ pairs, ignoring for now the role of $A$. In Section 4, sufficiency will be enforced by requiring the same pair to be feasible for every group.

---

[2]Some authors use "predictive parity" for equality of PPV alone; here we require equality of both PPV and FOR.

Let $Y$ be a binary target variable and denote by $\pi := P(Y = 1)$ its base rate, or prevalence, and let $S$ be a finite-valued calibrated prediction of $Y$. We denote the values in $\operatorname{supp} S$ in descending order, $1 \geq s_{\max} = s_1 > \cdots > s_m = s_{\min} \geq 0$, and write $P(s_i)$ for short when $S$ is implicitly understood. The calibration of $S$ implies that

$$\mathbb{E}[S] = \sum_{i \in [m]} P(s_i)\, s_i = \sum_{i \in [m]} P(s_i)\, P(Y = 1 | s_i) = \pi, \quad (1)$$

and to avoid degenerate cases, we assume that $m := |\operatorname{supp} S| > 1$, so $s_{\min} < \pi < s_{\max}$.

Let $R = R(S)$ be a (possibly randomized) binary classifier that predicts $Y$ based on the scores $S$ alone. More precisely, we consider a Markov chain $Y \leftrightarrow S \leftrightarrow R$, that is, $P(R|S, Y) = P(R|S)$. We restrict our analysis to nonconstant classifiers, and denote

$$p := \mathrm{PPV}(R) = P(Y = 1 \mid R = 1),$$
$$q := \mathrm{FOR}(R) = P(Y = 1 \mid R = 0).$$

The joint distribution of $S$ and $Y$ constrains the values of $p$ and $q$ that such classifiers can attain. For instance, the Markov condition together with the calibration of $S$ implies

$$p = \sum_{i \in [m]} P(Y = 1 \mid s_i)\, P(s_i \mid R = 1)$$
$$= \sum_{i \in [m]} s_i\, P(s_i \mid R = 1) \ \leq\ s_{\max}.$$

In this section, we characterize the set of feasible pairs $(p, q)$ attainable by classifiers of this form.

**Definition 3.1** (Feasibility). Given a joint distribution $P(Y, S)$, we say that a pair $(p, q) \in [0,1]^2$ is *feasible* if there exists a nonconstant randomized binary classifier $R = R(S)$, based on $S$ alone, such that $\mathrm{PPV}(R) = p$ and $\mathrm{FOR}(R) = q$. The *feasible region* is the set $\mathcal{C} = \mathcal{C}_{P(Y,S)} \subset [0,1]^2$ of all feasible pairs.

Since the labels of $R$ are arbitrary, there is symmetry between $p$ and $q$ (and thus in $\mathcal{C}$). Therefore, to simplify the analysis, we assume without loss of generality that $q \leq p$.

### 3.1. Geometry of the Feasible Region

The joint distribution of $R$ and $Y$ is completely determined by $p$, $q$ and the selection rate of $R$, $\mu := P(R = 1)$. Throughout, we restrict our attention to nonconstant classifiers, so $0 < \mu < 1$. Accordingly, we say that a pair $(p, q) \in \mathcal{C}$ is feasible with $\mu$ if it is attained by a classifier $R$ satisfying $P(R = 1) = \mu$. The base rate $\pi$ then enforces a linear constraint relating these three quantities, which admits a simple geometric interpretation.

If $(p, q) \in \mathcal{C}$ is feasible with $\mu$, then

$$\pi = P(R{=}1)P(Y{=}1|R{=}1) + P(R{=}0)P(Y{=}1|R{=}0)$$
$$= \mu p + (1 - \mu)q. \quad (2)$$

Geometrically, this implies that for any fixed $\mu$, all pairs $(p, q)$ feasible with $\mu$ lie on a single line of slope $-\mu/(1-\mu)$ passing through $(\pi, \pi)$. Moreover, since $q \leq p$ and $\mu > 0$, any nontrivial feasible pair satisfies $q < \pi < p$, with equality only when $p = q = \pi$.

The classifier $R$ itself is determined by its selection rule, that is, the selection probabilities $P(R = 1|s_i)$ for $i \in [m]$. Therefore, we can equivalently define feasibility in terms of selection rules by the following two equalities, in addition to (2):

$$\mu = \sum_{i \in [m]} P(s_i)\, P(R = 1|s_i) \quad (3)$$

and

$$p = \sum_{i \in [m]} P(Y = 1|s_i)\, P(s_i|R = 1)$$
$$= \sum_{i \in [m]} s_i\, \frac{P(s_i)}{P(R = 1)}\, P(R = 1|s_i)$$
$$= \frac{1}{\mu} \sum_{i \in [m]} s_i\, P(s_i)\, P(R = 1|s_i), \quad (4)$$

where the first equality follows from the Markov condition and the second from the calibration of $S$.

The linearity of (4) in the selection probabilities for a fixed $\mu$ implies another geometric property of $\mathcal{C}$, namely, its convexity at each given $\mu$. Consider a classifier $R^\eta$, defined by $P(R^\eta = 1|s_i) = \eta P(R = 1|s_i) + (1 - \eta)\mu$, for $\eta \in [0, 1]$. This is the $\eta$-mixture of $R$ with an independent Bernoulli random variable with parameter $\mu$, hence $P(R^\eta = 1) = \mu$ as well. Following the same steps as in (4), we have

$$\mathrm{PPV}(R^\eta) = \frac{1}{\mu} \sum_{i \in [m]} s_i\, P(s_i)\big(\eta P(R = 1|s_i) + (1 - \eta)\mu\big)$$
$$= \eta p + (1 - \eta) \sum_{i \in [m]} s_i\, P(s_i)$$
$$= \eta p + (1 - \eta)\pi,$$

where the last equality follows from (1). Similarly, algebraic manipulation using (2) shows that $\mathrm{FOR}(R^\eta) = \eta q + (1 - \eta)\pi$ (see Appendix A.1 for details). Therefore, the convex combination $\eta(p, q) + (1 - \eta)(\pi, \pi)$ is feasible with $\mu$ for all $\eta \in [0, 1]$.

This result does not imply that $\mathcal{C}$ is convex—in fact, it is typically not. However, it does imply that $\mathcal{C}$ is star-convex[3] with center at $(\pi, \pi)$. Since every feasible pair satisfies $p \geq \pi$ and $q \leq \pi$, it is therefore sufficient for the purpose of characterizing $\mathcal{C}$ to determine, for each fixed $\mu$, the extremal feasible values: the maximum achievable $p$ and the corresponding minimum $q$.

**Definition 3.2.** For $0 < \mu < 1$, let $\mathcal{C}_\mu \subset \mathcal{C}$ be the set of

---

[3]A set $\mathcal{S} \subseteq \mathbb{R}^d$ is *star-convex* with center $x_0 \in \mathcal{S}$ if, for every $x \in \mathcal{S}$, the line segment between $x_0$ and $x$ is contained in $\mathcal{S}$.

pairs $(p, q)$ feasible with $\mu$. Define

$$p^*(\mu) := \sup\{\, p \in [\pi, 1] \mid (p, q) \in \mathcal{C}_\mu \text{ for some } q \in [0, \pi] \,\},$$
$$q^*(\mu) := \inf\{\, q \in [0, \pi] \mid (p, q) \in \mathcal{C}_\mu \text{ for some } p \in [\pi, 1] \,\}.$$

For a fixed $\mu$, (3) and (4) show that maximizing $p$ over all selection rules $P(R = 1|S)$ is a linear program, equivalent to a fractional knapsack problem. The next theorem applies the greedy solution of this problem (Dantzig, 1957) to show that $(p^*(\mu), q^*(\mu))$ is feasible for all $0 < \mu < 1$, and is attained by a soft thresholding rule on $S$.

**Theorem 3.3.** *Let* $0 < \mu < 1$ *and define* $k^* = k^*(\mu) := \min\{\, k \mid \sum_{i \le k} P(s_i) \ge \mu \,\}$. *Then the pair* $(p^*(\mu), q^*(\mu))$ *is attained by* $R^* = R^*(S; \mu)$, *defined according to*

$$P(R^* = 1|s_i) = \begin{cases} 1 & i < k^*, \\ \dfrac{\mu - \sum_{j < k^*} P(s_j)}{P(s_{k^*})} & i = k^*, \\ 0 & i > k^*. \end{cases} \quad (5)$$

A detailed proof is given in Appendix B.

Since the scores $s_i$ are listed in descending order, the classifier $R^*$ acts almost like a threshold: it deterministically predicts 1 for all score bins strictly above $s_{k^*}$ and 0 for all score bins strictly below $s_{k^*}$, while possibly returning a randomized prediction on the bin corresponding to $s_{k^*}$. An important consequence of Theorem 3.3 is that $R^*(S; \mu)$ becomes a purely deterministic ("hard") threshold exactly at the values $\mu_k := \sum_{i \le k} P(s_i)$ for $k \in [m]$. In those cases we have $R^*(S; \mu_k) = 1$ iff $S \ge s_k$.

Moreover, Theorem 3.3, together with the convexity argument above, yields an explicit construction of a selection rule—and hence a classifier $R(S)$—to attain any given pair $(p, q) \in \mathcal{C}$. Specifically, such a classifier is obtained as an appropriate mixture of $R^*(\mu)$ and an independent Bernoulli random variable with parameter $\mu$, where $\mu = (\pi - q)/(p - q)$ follows from (2) (for $p, q \ne \pi$; otherwise the claim is trivial).

### 3.2. Boundary of the Feasible Region

As discussed above, the star-convexity of $\mathcal{C}$ implies that its structure is fully determined by its boundary. The boundary of $\mathcal{C}$ can therefore be divided into two parts. First, a *trivial* part consisting of two straight edges at $p = \pi$ and $q = \pi$, implied by the constraint $q \le \pi \le p$; these edges are not feasible (except at $(p, q) = (\pi, \pi)$), since we require $0 < \mu < 1$. Second, a *nontrivial* part, traced by $(p^*(\mu), q^*(\mu))$ as $\mu$ ranges over $(0, 1)$. In what follows, we refer to this nontrivial part simply as the boundary of $\mathcal{C}$ and denote it by $\partial \mathcal{C}$. The simple form of the selection rule in Theorem 3.3 allows for a piecewise characterization of $\partial \mathcal{C}$.

Recall the values $\mu_k := \sum_{i \le k} P(s_i)$, which correspond to the deterministic thresholds on $S$, and let $\mu_0 := 0$. We partition the interval $(0, 1)$ accordingly into $m$ subintervals of the form $I_k := (\mu_{k-1}, \mu_k] \cap (0, 1)$. Note that $k^*(\mu)$, as defined in Theorem 3.3—that is, the index of the score bin on which $R^*$ may randomize—is constant on each interval $I_k$ and equals $k$. Thus, combining Equation (5) with Equation (4), we obtain a simple explicit formula for $p^*(\mu)$ on each interval $I_k$:

$$p^*(\mu) = \frac{1}{\mu}\left(\sum_{i<k} s_i P(s_i) + s_k\left(\mu - \sum_{j<k} P(s_j)\right)\right)$$
$$= s_k + \frac{c_k}{\mu}, \quad (6)$$

where we define $c_k := \sum_{i<k} P(s_i)(s_i - s_k)$ for short.

One can show that $p^*(\mu)$ is continuous at the endpoints of the partition intervals $I_k$ (see Appendix A.2). Moreover, for $k = 1$ we have $c_1 = 0$, hence $p^*(\mu) \equiv s_1 = s_{\max}$ is constant on $I_1$. In contrast, for $k > 1$, we have $c_k > 0$, since the scores are in strictly descending order. Therefore, $p^*(\mu)$ is strictly decreasing in $\mu$ on each $I_k$ with $k > 1$, and, by continuity, it is strictly decreasing on $(\mu_1, 1)$.

In the following sections, in order to account for fairness constraints, we extend the analysis to multiple groups and study the intersection of their respective feasible regions. While parametrizing the boundary by the selection rate $\mu$ was convenient for deriving the preceding formulas, in the multi-group setting the group-wise selection rates may differ. It is therefore more convenient to use $p$ as a common boundary parameter. To this end, we exploit the monotonicity established above and induce a partition of the interval $(\pi, s_{\max})$ from the partition $\{I_k\}$ of $(0, 1)$.

For each $k > 1$, define $J_k := p^*(I_k)$. Then $J_k = [p_k, p_{k-1})$ for $1 < k < m$ and $J_m = (p_m, p_{m-1})$, where $p_k := p^*(\mu_k) = \frac{1}{\mu_k}\sum_{i \le k} s_i P(s_i)$ (see Appendix A.3 for details). By Equation (1), $p_m = \pi$, and since $p_1 = s_1 = s_{\max} > \pi$, the intervals $\{J_k\}$ form a partition of $(\pi, s_{\max})$. In particular, by the strict monotonicity of $p^*$ on $(\mu_1, 1)$, each $J_k$ satisfies $I_k = p^{*-1}(J_k)$.

Now, let $(p, q) \in \partial \mathcal{C}$ with $p < s_{\max}$, so $p \in J_k$ for some $k > 1$. Since the boundary consists of the points $(p^*(\mu), q^*(\mu))$, there exists $0 < \mu < 1$, such that $p = p^*(\mu)$, and it follows that $\mu \in p^{*-1}(J_k) = I_k$. Therefore, by Equation (6), $\mu = \mu(p) = c_k/(p - s_k)$. Plugging this into Equation (2) and rearranging the expression gives us a formula for $q^*$ in terms of $p$, instead of $\mu$:

$$q = q(p) = q^*(\mu(p)) = \frac{(\pi - c_k)p - s_k \pi}{p - s_k - c_k}. \quad (7)$$

This analysis brings us to the following characterization of the boundary of the feasible region:

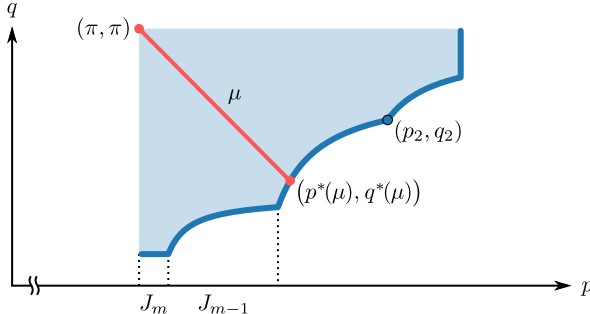

**Figure 1. Schematic of the feasible region $\mathcal{C}$.** The shaded region depicts a typical feasible region $\mathcal{C}$ corresponding to $m = 5$ distinct score values; the blue curve denotes its nontrivial boundary $\partial\mathcal{C}$. For a fixed selection rate $\mu$, all feasible pairs $(p, q)$ lie on the red line segment connecting $(\pi, \pi)$ to $(p^*(\mu), q^*(\mu))$. The boundary is piecewise over intervals $J_k$ on the $p$-axis; for illustration, only $J_m$ and $J_{m-1}$ are shown. Among the $m$ breakpoints of the boundary, the point $(p_2, q_2)$ is shown as an example and corresponds to the deterministic threshold classifier $R^*(S; \mu_2)$.

**Theorem 3.4.** *The closure of the nontrivial boundary curve of $\mathcal{C}$ consists of a continuous nondecreasing curve $q(p)$, defined piecewise by Equation* (7) *on each interval $J_k \subset [\pi, s_{\max})$, together with a vertical segment at $(s_{\max}, q)$ for $q \in [q(s_{\max}^-), \pi]$.*

The proof appears in Appendix C.[4]

More specifically, $\partial\mathcal{C}$ consists of two straight segments at $q = s_{\min}$ and $p = s_{\max}$, and, if $m > 2$, a sequence of hyperbolic arcs on each $J_k$ with $1 < k < m$ (see Figure 1). The breakpoints of this piecewise definition, $p_k$, correspond exactly to the values $\mu_k$, that is, to the hard threshold classifiers $R^*(S; \mu_k)$.

Finally, Theorem 3.4 implies the following bounds for $\mathcal{C}$: if $(p, q) \neq (\pi, \pi)$ is feasible with $0 < \mu < 1$, then

$$s_{\min} \le q^*(\mu) \le q < \pi < p \le p^*(\mu) \le s_{\max}. \quad (8)$$

## 4. Feasibility with Fairness Constraints

Having characterized the feasible region in the unconstrained setting, we now return to the original problem. In addition to a binary target variable $Y$ and a score $S$ representing a prediction of $Y$, the setting includes a finite-valued variable $A$ representing subgroups of the population. For simplicity, we assume that $A$ is binary; the extension to arbitrary finite support is straightforward. As a concrete example, $Y$ may indicate whether an individual completes a college degree, $S$ an admission test score, and $A$ a protected

---

[4] Formally, $\partial\mathcal{C}$ excludes the endpoints at $p = \pi$ and $q = \pi$, which correspond to degenerate classifiers with $\mu = 0$ and $\mu = 1$, respectively. Nonetheless, working with the closed boundary simplifies the subsequent analysis.

demographic attribute, such as gender.

In this setting, we assume that $S$ is a finite-valued group-calibrated prediction of $Y$, meaning that for all $s \in \operatorname{supp} S$ and $a \in \{0, 1\}$, $P(Y = 1|S = s, A = a) = s$. We build on the notation of the previous section, adding explicit reference to $A$ when needed. In particular, we write $\pi^a := P(Y = 1|a)$ and we denote by $\operatorname{supp}^a S := \{s \in \operatorname{supp} S \mid P(s|a) > 0\}$ the effective support of $S$ on the subgroup $a$. Importantly, we make no assumptions on these supports beyond finiteness: they may coincide across groups or be entirely disjoint. For each subgroup separately, as in the previous section, we denote $m^a := |\operatorname{supp}^a S|$ and list the distinct scores in $\operatorname{supp}^a S$ in decreasing order, $1 \ge s_{\max}^a = s_1^a > \cdots > s_{m^a}^a = s_{\min}^a \ge 0$.

We are interested in (possibly randomized) binary classifiers $R = R(S, A)$ that take both $S$ and $A$ as input. For each $a \in \{0, 1\}$, this implies the Markov condition $P(R|S, Y, A = a) = P(R|S, A = a)$, meaning that within each group, the classification depends on $S$ alone. For example, $R$ may represent a college admission decision based on an applicant's test score and protected attribute. We denote $\mu^a := P(R = 1|a)$, $p^a = \operatorname{PPV}^a(R) := P(Y = 1|R = 1, a)$, and $q^a = \operatorname{FOR}^a(R) := P(Y = 1|R = 0, a)$, and consider group-specific selection rules $P(R|s_i^a, a)$.

Together with the group-calibration of $S$, the group-wise Markov property of $R$ establishes a direct analogy with the single-group setting. Specifically, conditioning on a fixed group $A = a$ recovers exactly the same structural conditions as in the unconstrained case. As a result, feasibility can be analyzed separately within each subgroup.

**Definition 4.1** (Subgroup-Feasibility). Given a joint distribution $P(Y, S, A)$, we say that a pair $(p^a, q^a) \in [0, 1]^2$ is *subgroup-feasible* for $a \in \operatorname{supp} A$ if it is feasible in the sense of Definition 3.1, given the conditional distribution $P(Y, S|a)$. The *subgroup-feasible region* $\mathcal{C}^a := \mathcal{C}_{P(Y,S|A=a)}$ is defined accordingly.

### 4.1. Sufficiency

By Definition 2.1, $R$ satisfies sufficiency with respect to $Y$ and $A$ if $P(Y = 1|R, A) = P(Y = 1|R)$. In our setting, this is equivalent to requiring $p^0 = p^1 = p$ and $q^0 = q^1 = q$ (see Definition 2.3). Therefore, $R$ satisfies sufficiency if

$$(p, q) \in \mathcal{C}^0 \cap \mathcal{C}^1,$$

making the intersection of the subgroup-feasible regions our main object of interest.

By Equation (8), if $(p, q) \in \mathcal{C}^0 \cap \mathcal{C}^1$ then

$$\max\{\pi^0, \pi^1\} < p \le \min\{s_{\max}^0, s_{\max}^1\}$$
$$\text{and} \quad \max\{s_{\min}^0, s_{\min}^1\} \le q < \min\{\pi^0, \pi^1\}. \quad (9)$$

If these inequalities are infeasible or, more generally, if the intersection $\mathcal{C}^0 \cap \mathcal{C}^1$ is empty, there always exist classifiers $R = R(S, A)$ that satisfy sufficiency in a degenerate manner, meaning that they are constant on one or both of the subgroups. We treat degenerate cases in Appendix D.

## 4.2. The Intersection's Boundary

In what follows, we assume that $\mathcal{C}^0 \cap \mathcal{C}^1 \neq \varnothing$ and, without loss of generality, $\pi^0 \leq \pi^1$.

By Theorem 3.4, the boundary of each of the subgroup-feasible regions is nondecreasing in $p$. Therefore, the (non-trivial) boundary of the intersection, $\partial(\mathcal{C}^0 \cap \mathcal{C}^1)$, is given by the pointwise maximum of the two curves and is itself nondecreasing. More precisely, $\partial(\mathcal{C}^0 \cap \mathcal{C}^1)$ is defined by

$$q(p) = \max\{q^0(p), q^1(p)\}, \tag{10}$$

possibly including a vertical segment at $p = \min\{s^0_{\max}, s^1_{\max}\}$. On portions of $\partial(\mathcal{C}^0 \cap \mathcal{C}^1)$ where it coincides with the boundary of $\mathcal{C}^a$, we say that $\partial\mathcal{C}^a$ is the *active* boundary.

Before turning to a precise characterization of this curve, we make a few remarks that clarify its geometric and algorithmic implications.

First, every point on the boundary of $\mathcal{C}^0 \cap \mathcal{C}^1$ lies on the boundary of at least one of the subgroup-feasible regions $\mathcal{C}^0$ or $\mathcal{C}^1$, but not necessarily on both. As a consequence, in the typical case, almost all boundary points of $\mathcal{C}^0 \cap \mathcal{C}^1$ are interior points of one of the subgroup-feasible regions. In particular, this implies that in at least one subgroup such points cannot be attained by a deterministic (hard) threshold on $S$. Even more strikingly, the boundary of $\mathcal{C}^0 \cap \mathcal{C}^1$ may fail to contain *any* point corresponding to a deterministic threshold in either subgroup (see, for example, Figure 2).

Second, since $\partial(\mathcal{C}^0 \cap \mathcal{C}^1)$ is a nondecreasing curve, it completely determines the entire intersection and, consequently, all the binary classifiers $R(S, A)$ that satisfy sufficiency. Nevertheless, in many cases, as we show in the following sections, we are specifically interested in points lying on the boundary itself. Therefore, tracing this boundary is of interest in its own right. The piecewise definition of the boundary of each subgroup-feasible region, given by Equation (7), suggests a clean algorithmic approach, as we show next.

We now turn to a precise characterization of the closure of $\partial(\mathcal{C}^0 \cap \mathcal{C}^1)$, beginning with its effective domain. By the bounds in Equation (9), the representation of $q(p)$ in Equation (10) applies for $p \in [\pi^1, \min\{s^0_{\max}, s^1_{\max}\}]$, subject to the constraint $q(p) \leq \pi^0$. Since $q^0(p) \leq \pi^0$ for all $p$, the only potential restriction arises when $q^1(p)$ exceeds $\pi^0$.

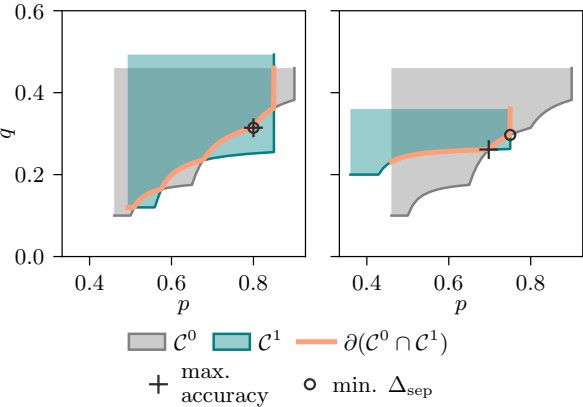

*Figure 2.* **Intersection of feasible regions.** Two examples of intersections between $\mathcal{C}^0$ (gray) and $\mathcal{C}^1$ (green); the orange curve denotes the boundary $\partial(\mathcal{C}^0 \cap \mathcal{C}^1)$ of the intersection. *Left:* the group-wise boundaries intersect four times. *Right:* the group-wise boundaries intersect twice; in particular, $\partial(\mathcal{C}^0 \cap \mathcal{C}^1)$ contains no breakpoints of either group, corresponding to deterministic threshold classifiers. In both cases, assuming $P(A = 1) = \frac{1}{2}$, the point on $\partial(\mathcal{C}^0 \cap \mathcal{C}^1)$ that maximizes overall accuracy (equivalently, minimizes the 0–1 loss) is marked by a black cross. The point that minimizes the deviation from separation, $\Delta_{\text{sep}}(R)$, is marked by a black circle. In the left example, the two optima coincide, whereas in the right example the loss- and deviation-minimizing points occur at distinct locations along the boundary.

Accordingly, we define

$$p^1_{\max} := \max\{p \leq s^1_{\max} \mid q^1(p) \leq \pi^0\} \tag{11}$$

(which exists because $\mathcal{C}^0 \cap \mathcal{C}^1 \neq \varnothing$ implies $s^1_{\min} < \pi^0$) and let $p_{\max} := \min\{s^0_{\max}, p^1_{\max}\}$. It follows that $q(p)$ is defined for all $p \in J := [\pi^1, p_{\max}]$. In Appendix E we provide an algorithm to calculate $p^1_{\max}$.

Recall the partitions $J^a_k = [p^a_k, p^a_{k-1})$, which define the piecewise structure of $q^a(p)$ for each $a \in \{0, 1\}$. We combine them to obtain a partition of $J$, defined by $J_{k,l} := J \cap J^0_k \cap J^1_l$ for all $1 < k \leq m^0$ and $1 < l \leq m^1$ for which the intersection is nonempty. On each subinterval $J_{k,l}$, the formulas for $q^0(p)$ and $q^1(p)$ are fixed, and given by (7). However, the active boundary on $J_{k,l}$, that is, $\arg\max_{a \in \{0,1\}} q^a(p)$, may change if the two curves cross. A simple rearrangement of Equation (7) shows that $q^0(p) \geq q^1(p)$ on $J_{k,l}$ iff $\Phi_{k,l}(p) \geq 0$, where $\Phi_{k,l}$ is a quadratic whose coefficients depend on $\pi^0, s^0_k, c^0_k$ and $\pi^1, s^1_l, c^1_l$. The exact expressions are given in Appendix F.

Since $\Phi_{k,l}(p)$ has at most two real roots in $J_{k,l}$, we can further partition it into subintervals $J_{k,l,i}$ (at most three), according to the sign of $\Phi_{k,l}$ in the interior. On each such subinterval, both the formulas for $q^0(p)$ and $q^1(p)$ and their relative ordering are fixed, and hence so is the active boundary. Consequently, the expression for $q(p)$ is fixed as well, being equal to $q^0(p)$ when $\Phi_{k,l} \geq 0$ in the interior of $J_{k,l,i}$,

**Algorithm 1 Boundary Trace.** Iterates over $\partial(\mathcal{C}^0 \cap \mathcal{C}^1)$, while keeping track of the active group and the indices $k$ and $l$, and evaluating a given objective function $\mathcal{F}$. See Appendix F for the definition of $\Phi_{k,l}$.

---

**Input:** Scores and weights $(s_i^a, P(s_i^a|A = a))$ for each of the groups $a \in \{0, 1\}$, with scores in descending order; group probabilities $P(a)$
**Objective:** Objective function $\mathcal{F}$

Compute $p_{\max}$ using Algorithm 2 (see Appendix E)
$p_L \leftarrow \max_{a \in \{0,1\}} \pi^a$
**while** $p_L < p_{\max}$ **do**
    $k \leftarrow \max\{k \in \{2, \ldots, m^0\} : p_{k-1}^0 > p_L\}$
    $l \leftarrow \max\{l \in \{2, \ldots, m^1\} : p_{l-1}^1 > p_L\}$
    $p_R \leftarrow \min\{p_{k-1}^0, p_{l-1}^1, p_{\max}\}$
                            // we have $J_{k,l} = [p_L, p_R]$
    $\text{roots}_\Phi \leftarrow$ real roots of $\Phi_{k,l}(p)$ in $(p_L, p_R)$, in increasing order
    **for each** $p_R' \in \text{roots}_\Phi \cup \{p_R\}$ **do**
                            // and now $J_{k,l,i} = [p_L, p_R']$
        $a \leftarrow 0$ **if** $\Phi_{k,l}\left(\frac{p_L + p_R'}{2}\right) \geq 0$ **else** $a \leftarrow 1$
                            // $a$ is the active boundary on $J_{k,l,i}$
        Evaluate $\mathcal{F}$ on $[p_L, p_R']$ using $a, k, l$
        $p_L \leftarrow p_R'$
    **end for**
**end while**
**if** $p_{\max} = \min\{s_{\max}^0, s_{\max}^1\}$ **then**
    Evaluate $\mathcal{F}$ on the vertical segment at $p = p_{\max}$
**end if**

---

and to $q^1(p)$ otherwise.

Iterating over all nonempty intervals $J_{k,l,i}$ returns the boundary curve of $\mathcal{C}^0 \cap \mathcal{C}^1$, unless $p_{\max} = \min\{s_{\max}^0, s_{\max}^1\}$. In that case, it remains to trace the vertical segment at $p = p_{\max}$, given by $(p_{\max}, q)$ for $q \in [q(p_{\max}^-), \pi^0]$.

In Algorithm 1 we summarize the complete procedure for tracing $\partial(\mathcal{C}^0 \cap \mathcal{C}^1)$, while computing an arbitrary objective function along the boundary. As we show next, this approach can be used to search for a point on the boundary that minimizes a given loss function (see Section 5) or the deviation from separation (see Section 6).

## 5. Loss Minimization Under Sufficiency

Up to this point, we have characterized all pairs $(p, q)$ attainable as $p = \text{PPV}(R) := P(Y = 1 \mid R = 1)$ and $q = \text{FOR}(R) := P(Y = 1 \mid R = 0)$ by binary classifiers $R = R(S, A)$ that satisfy predictive parity. Specifically, for such classifiers the group-wise quantities $\text{PPV}^a(R) := P(Y = 1 \mid R = 1, a)$ and $\text{FOR}^a(R) := P(Y = 1 \mid R = 0, a)$ coincide with $p$ and $q$ for all groups $a$. Moreover, Theorem 3.3 together with the star-convexity

property yields an explicit description of the corresponding selection rules $P(R = 1 \mid S, A)$ for each feasible pair $(p, q)$ in the intersection of the group-specific feasible regions.

When this intersection is nonempty, a natural remaining question is how to choose among the feasible classifiers. In this section, we address this question by considering classifiers that minimize a given loss function.

Let $\ell : \{0, 1\}^2 \to \mathbb{R}$ be a loss function assigning a cost $\ell(R, Y)$ to a prediction $R$ and a true label $Y$, and write $\ell_{ry} := \ell(r, y)$. Without loss of generality, we assume $\ell_{00} = \ell_{11} = 0$ and $\ell_{01} + \ell_{10} > 0$ (otherwise, the labels of $R$ can be flipped). Our goal is to find a classifier $R$ that minimizes the expected loss $L := \mathbb{E}_{R,Y}[\ell(R, Y)]$ subject to sufficiency. We show that any optimal classifier attains a pair $(p, q) \in \mathcal{C}^0 \cap \mathcal{C}^1$ lying on the boundary of the intersection, and can therefore be found by tracing this boundary using Algorithm 1.

For each $a \in \{0, 1\}$ we have

$$
\begin{aligned}
\mathbb{E}[\ell(R, Y)|a] &= P(R = 1|a) \, P(Y = 0|R = 1, a)\ell_{10} \\
&\quad + P(R = 0|a) \, P(Y = 1|R = 0, a)\ell_{01} \\
&= \mu^a(1 - p^a)\ell_{10} + (1 - \mu^a)q^a\ell_{01} \\
&= \pi^a\ell_{01} + \mu^a\ell_{10} - \mu^a p^a(\ell_{01} + \ell_{10}),
\end{aligned}
$$

where the last equality follows from (2).

Define the aggregate parameters $\pi := P(Y = 1) = P(A = 0)\pi^0 + P(A = 1)\pi^1$ and $\mu := P(R = 1) = P(A = 0)\mu^0 + P(A = 1)\mu^1$. Under sufficiency, that is, when $p^0 = p^1 = p$ and $q^0 = q^1 = q$, we get

$$
L = \mathbb{E}\left[\, \mathbb{E}[\ell(R, Y)|A]\right] = \pi\ell_{01} + \mu\ell_{10} - \mu p(\ell_{01} + \ell_{10}). \tag{12}
$$

In addition, as in the unconstrained setting, $\pi = \mu p + (1 - \mu)q$, so for a fixed $\mu$, all feasible pairs $(p, q)$ attained by classifiers with $P(R = 1) = \mu$ lie on the same line that passes through $(\pi, \pi)$. Since $\ell_{01} + \ell_{10} > 0$, (12) implies that, for any fixed $\mu$, the expected loss is minimized by maximizing $p$. Consequently, the loss-minimizing pair $(p, q)$ must lie on the boundary of the feasible set, which under sufficiency is precisely $\partial(\mathcal{C}^0 \cap \mathcal{C}^1)$.

Finally, for any point $(\pi, \pi) \neq (p, q) \in \mathcal{C}^0 \cap \mathcal{C}^1$, Equation (2) gives $\mu = (\pi - q)/(p - q)$. Substituting this expression into (12) yields

$$
L = \pi\ell_{01} + \frac{\pi - q}{p - q}\big(\ell_{10} - p(\ell_{01} + \ell_{10})\big). \tag{13}
$$

Using Algorithm 1 to trace the boundary of $\mathcal{C}^0 \cap \mathcal{C}^1$, we can search for the exact minimizer of (13) on each interval $J_{k,l,i}$ (and the vertical segment at $p = p_{\max}$, if it exists). As shown in Appendix G, this reduces to solving a quadratic equation and evaluating (13) at the interval endpoints, as well as at any interior critical points.

## 6. Minimizing Deviation from Separation

Sufficiency and separation (equalized odds) are known to be incompatible, except in trivial cases (Chouldechova, 2017; Kleinberg et al., 2017). Nevertheless, among classifiers that satisfy sufficiency, the extent to which separation is violated can vary. In this section, we adopt the deviation-from-separation measure $\Delta_{\text{sep}}(R)$ introduced in (Benger & Ligett, 2025), and study classifiers that satisfy predictive parity while minimizing $\Delta_{\text{sep}}(R)$. We show that the optimal classifier is attained on the boundary of $\mathcal{C}^0 \cap \mathcal{C}^1$, and can therefore be found using Algorithm 1.

There is no guarantee that a classifier minimizing $\Delta_{\text{sep}}(R)$ is also optimal with respect to any particular loss function. Nevertheless, in practice we find that its accuracy is often comparable to that of the optimal classifier. Importantly, this analysis enables practitioners to explicitly compare the loss-optimal solution with a classifier that minimizes deviation from separation, and to select between these two different objectives based on application-specific considerations.

The deviation from separation, measured using total variation divergence,[5] is defined as

$$\Delta_{\text{sep}}(R) := \mathbb{E}_{A,Y} \, \text{TV} \big( P(R|A,Y), \, P(R|Y) \big).$$

This quantity measures the average discrepancy, across subgroups, between the subgroup-specific and aggregate true positive and false positive rates. In particular, separation—that is, in the binary case, equalized odds—requires these rates to coincide across subgroups, and hence any classifier satisfying equalized odds has $\Delta_{\text{sep}}(R) = 0$.

In Appendix H we show that

$$\Delta_{\text{sep}}(R) = K \left( \frac{1-\mu}{p-\pi} - \frac{\mu(p-\pi)}{\pi(1-\pi)} \right), \qquad (14)$$

where $K := 2P(A=1)\,P(A=0)\,|\pi^1 - \pi^0| \geq 0$, and $\mu$ and $\pi$ are the aggregate parameters defined above. Differentiating with respect to $p$ while holding $\mu$ fixed yields

$$\frac{d}{dp}\Delta_{\text{sep}}(R) = K \left( -\frac{1-\mu}{(p-\pi)^2} - \frac{\mu}{\pi(1-\pi)} \right) \leq 0,$$

which implies that, for any fixed $\mu$, the deviation from separation is either constant or minimized by maximizing $p$. Consequently, a feasible pair $(p,q)$ minimizing $\Delta_{\text{sep}}(R)$ must lie on the boundary of $\mathcal{C}^0 \cap \mathcal{C}^1$.

As in the loss-minimization case, Algorithm 1 can be used to search for the exact minimizer of (14) along the boundary. Appendix H shows that this reduces to solving a quadratic equation and evaluating $\Delta_{\text{sep}}(R)$ at the endpoints of each interval $J_{k,l,i}$, as well as at any interior critical points.

---

[5]$\text{TV}(P,Q) := \frac{1}{2}\sum_x |P(x) - Q(x)|.$

## 7. Experiments

We illustrate our method on three empirical case studies. Full implementation details and additional results are provided in Appendix I.

### 7.1. FICO

FICO scores are used to predict creditworthiness. We use the aggregate score statistics from Barocas et al. (2023), with $Y$ indicating non-default and race as the protected attribute $A$. We restrict attention to the "white" ($A = 0$) and "black" ($A = 1$) groups.

An unconstrained group-aware Bayes classifier based on these calibrated scores attains accuracy 0.8819, but violates predictive parity: the PPV values are 0.91 and 0.79, and the FOR values are 0.20 and 0.13, for the "white" and "black" groups, respectively. Our algorithm returns a classifier with common PPV 0.91, common FOR 0.23, and accuracy 0.8676. Figure 3 shows the feasible regions and the optimal point on the intersection boundary. The optimum lies on $\partial \mathcal{C}^1$ but in the interior of $\mathcal{C}^0$, so the corresponding rule for the "white" group cannot be realized by thresholding alone.

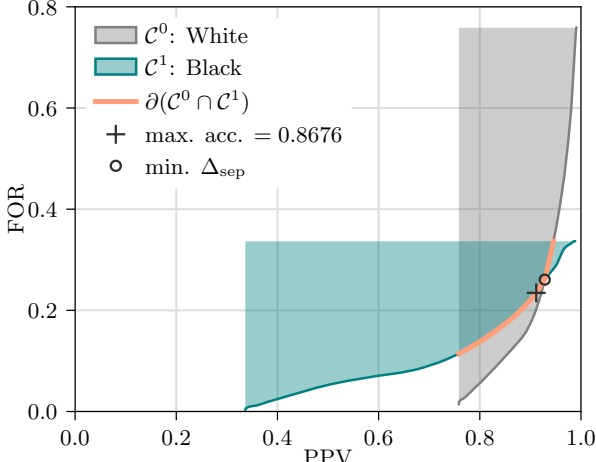

*Figure 3.* **FICO Scores.** Feasible regions for the "white" and "black" groups. With 200 score bins, the boundaries appear nearly smooth.

### 7.2. COMPAS

COMPAS scores are used to predict recidivism risk. We use the data collected by Angwin et al. (2016), with race as the protected attribute, and restrict attention to the "white" ($A = 0$) and "black" ($A = 1$) groups. Unlike in the FICO case, official aggregate score statistics are not available. Motivated by the approximate sufficiency of COMPAS scores (Chouldechova, 2017), we assign each decile score $d$

its pooled empirical value $P(Y = 1 \mid d)$, ignoring $A$, and estimate the group-specific weights $P(d \mid A = a)$. Thus, for this illustrative example, we treat the resulting scores as group-calibrated.

Figure 4 shows the feasible regions and the most accurate classifier satisfying sufficiency. The optimum lies on a breakpoint of $\partial \mathcal{C}^1$, corresponding to a deterministic threshold that selects all individuals with score at least 5 in the "black" group. In contrast, it lies in the interior of $\mathcal{C}^0$, so the optimal rule for the "white" group cannot be realized by thresholding alone. The nearest breakpoint of $\partial \mathcal{C}^0$ corresponds to the threshold selecting all individuals with score at least 6.

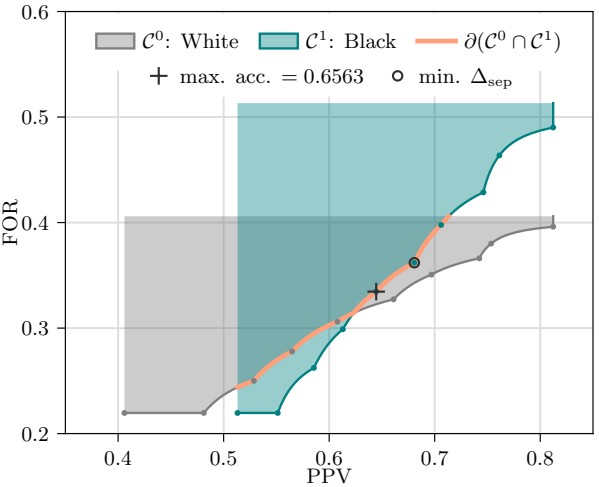

*Figure 4.* **COMPAS Scores.** Feasible regions for the "white" and "black" groups. Dots on the boundaries denote breakpoints corresponding to deterministic thresholds on the 10 decile scores.

### 7.3. ACS Income

Finally, we provide an end-to-end implementation of our method on the California subset of the ACS Income dataset, based on the American Community Survey of the US Census Bureau (Ding et al., 2021). Here group-calibrated scores are not given directly. The data contain demographic, education, and occupation features, and the task is to predict whether income exceeds a fixed threshold. We use sex as the protected attribute, with $A = 0$ for the "male" group and $A = 1$ for the "female" group.

In each repetition, we train a logistic regression model, use out-of-fold training predictions to calibrate its outputs separately for each group using $k = 100$ equal-mass bins, and apply our post-processing algorithm on a held-out test set. The binning and calibrated score values are estimated from these out-of-fold predictions, while the score-bin weights are estimated on the test set without using test labels. We repeat this procedure over 100 train/test splits.

Because calibration is estimated from out-of-fold training predictions, the scores are not perfectly group-calibrated on the held-out test set, and the resulting classifier need not satisfy exact predictive parity there. Nevertheless, the induced violations are small: the mean PPV gap $|\operatorname{PPV}^0(R) - \operatorname{PPV}^1(R)|$ is $4.8 \times 10^{-3}$, and the mean FOR gap $|\operatorname{FOR}^0(R) - \operatorname{FOR}^1(R)|$ is $3.7 \times 10^{-3}$. The post-processed classifier attains accuracy 0.8167, compared with 0.8190 for the unconstrained classifier.

## 8. Conclusion

Unlike independence and separation, sufficiency is not generally preserved under post-processing. This is a major concern when predicted probabilities are converted into binary classifications or decisions: even if the underlying predictions satisfy perfect sufficiency, applying a simple threshold may result in unfair decisions. In this work, we give an exact solution for optimal binary classification under sufficiency, assuming finite group-calibrated scores. We provide a geometric characterization of the feasible pairs of positive predictive value (PPV) and false omission rate (FOR) attainable by such classifiers, and use it to derive a simple post-processing algorithm to attain the optimal classifier with respect to various objectives. Importantly, once group-calibrated scores and group membership are available, our method produces fair decision rules without access to raw features, additional labels, or retraining a predictor.

The assumption of group-calibrated scores is weaker than assuming access to the true conditional label probabilities. In practice, such scores may be obtained by standard post-hoc calibration methods (see, e.g., Niculescu-Mizil & Caruana, 2005; Guo et al., 2017) or through stronger notions such as multicalibration (Hébert-Johnson et al., 2018). The finite-score setting is also natural in light of the connection between distribution-free calibration and discretization into finitely many score bins (Gupta et al., 2020). Since calibration is only approximate in finite samples, we provide a robustness analysis for calibration error (Appendix J); empirically, the induced violations of predictive parity remain small, and often much smaller than the worst-case bound.

Finally, the geometric view highlights a limitation of exact sufficiency for binary classification with multiple protected groups. A sufficient classifier exists only when all the group-specific feasible regions have a nonempty intersection. As the number of groups increases, this requirement becomes more restrictive, and exact feasibility may fail more often in practice. This suggests that relaxing the strict sufficiency criterion and seeking group-specific rules that minimize the differences of PPV and FOR across more than two groups, even when those quantities cannot achieve equality, may be a promising direction for future work.

## Acknowledgments

This work was supported in part by ERC grant 101125913, Simons Foundation Collaboration 733792, Apple, and a grant from the Israeli Council of Higher Education. Views and opinions expressed are however those of the authors only and do not necessarily reflect those of the European Union or the European Research Council Executive Agency. Neither the European Union nor the granting authority can be held responsible for them.

## Impact Statement

This paper presents work whose goal is to advance fairness in data-driven decision-making. While there are many aspects of just decision-making that are not covered by this work, by giving an algorithm to find the optimal (randomized) binary classifier that satisfies sufficiency on the basis of a calibrated score, we give decision-makers new tools to strike better balances between accuracy and other societal desiderata.

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

# A. Additional Derivations

## A.1. Mixture FOR

Using (2), we have

$$
\begin{aligned}
(1 - \mu)\operatorname{FOR}(R^\eta) &= \pi - \mu\operatorname{PPV}(R^\eta) \\
&= \pi - \mu\big(\eta p + (1 - \eta)\pi\big) \\
&= (1 - \mu)\pi - \eta\mu p + \eta\mu\pi \\
&\overset{*}{=} (1 - \mu)\pi - \eta\big(\pi - (1 - \mu)q\big) + \eta\mu\pi \\
&= (1 - \mu)\pi - \eta(1 - \mu)\pi + \eta(1 - \mu)q \\
&= (1 - \eta)(1 - \mu)\pi + \eta(1 - \mu)q,
\end{aligned}
$$

where $(*)$ follows from (2) applied to $R$, namely $\mu p = \pi - (1 - \mu)q$. Dividing both sides by $(1 - \mu)$ yields

$$
\operatorname{FOR}(R^\eta) = \eta q + (1 - \eta)\pi.
$$

## A.2. Continuity of $p^*(\mu)$

To prove the continuity of $p^*(\mu)$ on the entire interval $(0, 1)$, it suffices to verify right continuity at the left endpoints of the subintervals $I_k = (\mu_{k-1}, \mu_k]$ for $k > 1$. Let $\mu \in I_{k+1}$ for some $k < m$, then by (6),

$$
\begin{aligned}
p^*(\mu) &= s_{k+1} + \frac{1}{\mu}\sum_{i \le k} P(s_i)(s_i - s_{k+1}) \\
&= s_{k+1} + \frac{1}{\mu}\sum_{i \le k} P(s_i)(s_k - s_{k+1}) + \frac{1}{\mu}\sum_{i \le k} P(s_i)(s_i - s_k) \\
&= s_{k+1} + \frac{\mu_k}{\mu}(s_k - s_{k+1}) + \frac{c_k}{\mu}.
\end{aligned}
$$

Consequently,

$$
p^*(\mu) \xrightarrow[\mu \to \mu_k^+]{} s_k + \frac{c_k}{\mu_k} = p^*(\mu_k).
$$

## A.3. Formula of $p(\mu_k)$

Using (6) and the definition $\mu_k := \sum_{i \le k} P(s_i)$, we get for all $k \in [m]$

$$
\begin{aligned}
p_k := p^*(\mu_k) &= s_k + \frac{c_k}{\mu_k} \\
&= s_k + \frac{\sum_{i < k} P(s_i)(s_i - s_k)}{\sum_{i \le k} P(s_i)} \\
&= \frac{\sum_{i < k} s_i P(s_i) + s_k \left(\sum_{i \le k} P(s_i) - \sum_{i < k} P(s_i)\right)}{\sum_{i \le k} P(s_i)} \\
&= \frac{\sum_{i \le k} s_i P(s_i)}{\sum_{i \le k} P(s_i)}.
\end{aligned}
$$

## A.4. Formula for $q(p)$

Let $\mu \in I_k$ with $k > 1$. By (6), $p = p(\mu) = s_k + \frac{c_k}{\mu} > s_k$, since $c_k > 0$. Inverting this relation yields

$$
\mu = \frac{c_k}{p - s_k}. \tag{15}
$$

Now, substituting into (2) gives

$$
\pi = \mu p + (1 - \mu)q = \frac{c_k p + (p - s_k - c_k)q}{p - s_k},
$$

so that

$$(p - s_k - c_k)q = (\pi - c_k)p - s_k\pi.$$

Therefore,

$$q = \frac{(\pi - c_k)p - s_k\pi}{p - s_k - c_k},$$

which is well defined, since $\mu < 1$ implies $p - s_k > c_k$ by (15).

## B. Proof of Theorem 3.3

**Theorem 3.3** (Restated). *Let* $0 < \mu < 1$ *and define* $k^* = k^*(\mu) := \min\{ k \mid \sum_{i \leq k} P(s_i) \geq \mu \}$. *Then the pair* $(p^*(\mu), q^*(\mu))$ *is attained by* $R^* = R^*(S; \mu)$, *defined according to*

$$P(R^* = 1|s_i) = \begin{cases} 1 & i < k^*, \\ \dfrac{1}{P(s_{k^*})} \left( \mu - \sum_{j < k^*} P(s_j) \right) & i = k^*, \\ 0 & i > k^*. \end{cases} \tag{16}$$

*Proof.* From Equations (3) and (4) we note that the problem of finding a selection rule that maximizes $p$ for a given $\mu$ is equivalent to the fractional knapsack problem with the constraints $\sum_{i \in [m]} x_i P(s_i) = \mu$ and $x_i \in [0, 1]$ for all $i \in [m]$, and the maximization objective $\sum_{i \in [m]} x_i v_i$, where the item values are given by $v_i := s_i P(s_i)$. The selection rule $P(R^*|s_i)$ is exactly the greedy algorithm's solution to this problem (Dantzig, 1957), as we assumed the scores are in descending order. Finally, from (2), it follows that for any fixed $0 < \mu < 1$, $q$ is a strictly decreasing function of $p$, meaning that $q^*(\mu)$ must be attained together with $p^*(\mu)$. $\square$

## C. Proof of Theorem 3.4

**Theorem 3.4** (restated). *The closure of the nontrivial boundary curve of $\mathcal{C}$ consists of a continuous nondecreasing curve $q(p)$, defined piecewise by (7) on each interval $J_k \subset [\pi, s_{\max})$, together with a vertical segment at $(s_{\max}, q)$ for $q \in [q(s_{\max}^-), \pi]$.*

*Proof.* Let $(p, q)$ lie on the nontrivial boundary of $\mathcal{C}$. Then there exists $\mu \in (0, 1)$ such that $(p, q) = (p^*(\mu), q^*(\mu))$.

If $\mu \in I_1 = (0, \mu_1]$, then $c_1 = 0$ and (6) gives $p = s_1 = s_{\max}$. Substituting into (2) and rearranging yields

$$q = q^*(\mu) = \frac{\pi - \mu s_{\max}}{1 - \mu}.$$

Differentiating with respect to $\mu$ gives

$$\frac{d}{d\mu}q = \frac{-s_{\max}(1 - \mu) + \pi - \mu s_{\max}}{(1 - \mu)^2} = \frac{\pi - s_{\max}}{(1 - \mu)^2}.$$

Since $m > 1$ and $\mathbb{E}[S] = \pi$ by (1), we have $s_{\max} > \pi$, and thus $q$ is strictly decreasing in $\mu$ on $I_1$. At the endpoints,

$$q^*(0^+) = \frac{\pi - 0 \cdot s_{\max}}{1 - 0} = \pi, \qquad\qquad q^*(\mu_1) = \frac{\pi - P(s_1) s_{\max}}{1 - P(s_1)} =: q_1,$$

where $q_1 < \pi$ since $P(s_1) > 0$ and $s_{\max} > \pi$. Hence the closure of the nontrivial boundary includes the vertical segment $(s_{\max}, q)$ for $q \in [q_1, \pi]$.

If $\mu \in I_k$ for some $k > 1$, then $p \in J_k$ by construction, and (7) gives

$$q = q(p) = \frac{(\pi - c_k)p - s_k\pi}{p - s_k - c_k}.$$

Differentiating with respect to $p$ yields

$$\frac{d}{dp}q(p) = \frac{(\pi - c_k)(p - s_k - c_k) - (\pi - c_k)p + s_k\pi}{(p - s_k - c_k)^2} = \frac{-c_k(\pi - s_k - c_k)}{(p - s_k - c_k)^2}, \tag{17}$$

so the sign of $\frac{d}{dp}q(p)$ on $J_k$ is determined by $\pi - s_k - c_k$. Now, for all $k \in [m]$,

$$
\begin{aligned}
\pi - s_k - c_k &= \sum_{i \in [m]} s_i P(s_i) - s_k - \sum_{i<k} P(s_i)(s_i - s_k) \\
&= \sum_{i \in [m]} P(s_i)(s_i - s_k) - \sum_{i<k} P(s_i)(s_i - s_k) \\
&= \sum_{i \geq k} P(s_i)(s_i - s_k),
\end{aligned}
\tag{18}
$$

and since $s_i < s_k$ for all $i > k$, it follows that $\pi - s_k - c_k < 0$ for all $k < m$, while $\pi - s_m - c_m = 0$. Because $c_k > 0$ for $k > 1$, (17) implies that $q(p)$ is strictly increasing on $J_k$ for $1 < k < m$, and constant on $J_m$.

In particular, $c_m = \pi - s_m$, so for $p \in J_m$,

$$
q(p) = \frac{p(\pi - \pi + s_m) - s_m \pi}{p - s_m - \pi + s_m} = \frac{s_m(p - \pi)}{p - \pi} = s_m = s_{\min}.
$$

Thus, although (7) is not defined at the left endpoint $p = \pi$, $q(p)$ extends continuously there by setting $q(\pi) := s_{\min}$.

Since the intervals $\{J_k\}$ cover $(\pi, s_{\max})$ and $q(p)$ matches continuously at their endpoints (by continuity of $p^*$ in $\mu$ and (2)), we conclude that $q(p)$ is nondecreasing on $[\pi, s_{\max})$ and attains its minimum $s_{\min}$ on the closure of $J_m$.

Finally, we verify that the graph of $q(p)$ on $[\pi, s_{\max})$ connects continuously to the vertical segment at $p = s_{\max}$. For $p$ sufficiently close to $s_{\max}$ from the left, we have $p \in J_2 = [p_2, s_{\max})$. On this interval, $c_2 = P(s_1)(s_1 - s_2) = P(s_1)(s_{\max} - s_2)$, and hence by (7),

$$
\begin{aligned}
q(s_{\max}^-) &= \frac{s_{\max}(\pi - c_2) - s_2 \pi}{s_{\max} - s_2 - c_2} \\
&= \frac{\pi(s_{\max} - s_2) - s_{\max} P(s_1)(s_{\max} - s_2)}{(s_{\max} - s_2) - P(s_1)(s_{\max} - s_2)} \\
&= \frac{\pi - P(s_1) s_{\max}}{1 - P(s_1)} = q_1,
\end{aligned}
$$

which is the lower endpoint of the vertical segment identified above. $\qquad\square$

## D. Degenerate Cases

Throughout this paper we restrict attention to nonconstant binary classifiers $R$, so their selection rate $\mu := P(R = 1)$ lies strictly between 0 and 1. When $R$ is constant, at least one of $\mathrm{PPV}(R) := P(Y = 1 | R = 1)$ or $\mathrm{FOR}(R) := P(Y = 1 | R = 0)$ is undefined. The same issue arises at the subgroup level: if $R$ is constant within a group $a$, then the corresponding $\mathrm{PPV}^a(R)$ or $\mathrm{FOR}^a(R)$ is undefined, and predictive parity cannot be stated for that group.

By contrast, the fairness criterion of sufficiency requires the conditional independence of $Y$ and $A$ given $R$. When both $Y$ and $R$ are binary and $R$ is nonconstant, this condition is equivalent to predictive parity, but it remains well defined even in degenerate cases where $R$ is constant (either overall or within a subgroup), since it constrains the distribution of $Y$ only on $(A, R)$ pairs with positive probability. Concretely, sufficiency requires

$$
P(Y = 1 | R = r, A = a) = P(Y = 1 | R = r)
\tag{19}
$$

for all $r$ and $a$ such that $P(R = r, A = a) > 0$.

While the utility of constant classification is debatable, one could still consider constant behavior on part of the population, provided the fairness constraint is satisfied. We now briefly characterize how such degenerate cases fit into our framework.

First, we extend the definition of feasibility to include constant classifiers:

**Definition D.1** (Feasibility with degenerate classifiers). Given a joint distribution $P(Y, S)$, we say that a pair $(p, q) \in [0, 1]^2$ is feasible if there exists a randomized binary classifier $R = R(S)$ with selection rate $\mu := P(R = 1) \in [0, 1]$, such that $P(Y = 1, R = 1) = \mu p$ and $P(Y = 1, R = 0) = (1 - \mu)q$.

Note that when $0 < \mu < 1$, the definition above reduces to Definition 3.1, since $\text{PPV}(R) = \frac{1}{\mu}P(Y = 1, R = 1)$ and $\text{FOR}(R) = \frac{1}{1-\mu}P(Y = 1, R = 0)$.

The identity $\pi = \mu p + (1 - \mu)q$ (2) remains valid even in the extreme cases $\mu \in \{0, 1\}$. Therefore, relying on the standing convention that $q \le \pi \le p$, we obtain that if $\mu = 0$, then $q = \pi$ and $p \in [\pi, 1]$ is arbitrary, whereas if $\mu = 1$, then $p = \pi$ and $q \in [0, \pi]$ is arbitrary. Geometrically, this implies that allowing degenerate classifiers adds two boundary segments to the feasible region, extending horizontally and vertically from the point $(\pi, \pi)$.

If $m = 1$, meaning that $S$ is constant, then any classifier $R = R(S)$ is constant as well, and the feasible region consists only of these degenerate edges. If $m > 1$, however, Theorem 3.4 guarantees the feasibility of the point $(s_{\max}, q(s_{\max}^-))$, where $s_{\max} > \pi$ and $q(s_{\max}^-) = \frac{\pi - P(s_1)s_{\max}}{1 - P(s_1)} < \pi$ (see Appendix C). This point does not lie on either degenerate segment, and in fact the structure of $\partial\mathcal{C}$ described in Theorem 3.4 implies that $\mathcal{C}$ has a nonempty interior.

When considering sufficiency with degenerate classifiers, we must therefore include the degenerate edges in the intersection analysis. For example, using the notation of Section 4, suppose that $(p, q) \in \mathcal{C}^0 \cap \mathcal{C}^1$ lies on the horizontal degenerate edge of $\mathcal{C}^0$ but in the interior of $\mathcal{C}^1$, so that $\mu^0 = 0$ and $0 < \mu^1 < 1$. Then $P(R = 1, A = 0) = 0$, but conditioning on $R = 0$ is valid, and by Definition D.1 we have

$$P(Y = 1|R = 0, A = 0) = \frac{P(Y = 1, R = 0|A = 0)}{P(R = 0|A = 0)} = \frac{(1 - \mu^0)q}{1 - \mu^0} = q,$$

$$P(Y = 1|R = 0, A = 1) = \frac{P(Y = 1, R = 0|A = 1)}{P(R = 0|A = 1)} = \frac{(1 - \mu^1)q}{1 - \mu^1} = q.$$

Hence $P(Y = 1|R = 0, A = a) = P(Y = 1|R = 0) = q$ for both groups, and sufficiency is satisfied by (19).

By (8), an intersection between the nontrivial part of $\mathcal{C}^a$ and a degenerate edge of $\mathcal{C}^b$ occurs if $s_{\min}^a \le \pi^b \le s_{\max}^a$ and $\pi^a \ne \pi^b$. For instance, if $s_{\min}^1 \le \pi^0 < \pi^1$, then the horizontal degenerate edge of $\mathcal{C}^0$ intersects $\mathcal{C}^1$ along the segment $(p, \pi^0)$ for $p \in [\pi^1, p_{\max}^1]$, where $p_{\max}^1$ is defined as in (11).

Finally, note that once degenerate classifiers are allowed, the intersection $\mathcal{C}^0 \cap \mathcal{C}^1$ is always nonempty, albeit in a trivial way. Assuming $\pi^0 < \pi^1$, the point $(\pi^1, \pi^0)$ lies simultaneously on the horizontal degenerate edge of $\mathcal{C}^0$ and the vertical degenerate edge of $\mathcal{C}^1$. The classifier attaining this point is $R = A$, which simply predicts the subgroup identity. Interpreted probabilistically, this corresponds to predicting the subgroup base rate for every individual.

## E. Computing $p_{\max}$ and $q_{\min}$

Assume without loss of generality that $\pi^0 \le \pi^1$ and that $\mathcal{C}^0 \cap \mathcal{C}^1 \ne \varnothing$. Then (9) implies that $s_{\min}^1 < \pi^0$ and $\pi^1 < s_{\max}^0$. By Theorem 3.4, the endpoints of the boundary curve of $\mathcal{C}^1$ are $(\pi^1, s_{\min}^1)$ and $(s_{\max}^1, \pi^1)$, hence it must cross (or at least attain) the value $q = \pi^0$ at some $p \in [\pi^1, s_{\max}^1]$.

Define $p_{\max}^1$ according to (11), namely

$$p_{\max}^1 := \max\{p \in [\pi^1, s_{\max}^1] \mid q^1(p) \le \pi^0\}.$$

This quantity is well defined by the argument above. Moreover, any $(p, q) \in \mathcal{C}^1$ with $p > p_{\max}^1$ must satisfy $q > \pi^0$, and therefore, by (9), does not belong to the intersection $\mathcal{C}^0 \cap \mathcal{C}^1$. Consequently, for all $(p, q) \in \mathcal{C}^0 \cap \mathcal{C}^1$ we have $p \le p_{\max} := \min\{s_{\max}^0, p_{\max}^1\}$, with equality attained on $\partial\mathcal{C}^0$ if $p_{\max} = s_{\max}^0$, and on $\partial\mathcal{C}^1$ if $p_{\max} = p_{\max}^1$.

We next provide an algorithmic way to compute $p_{\max}^1$, based on the piecewise characterization of $\partial\mathcal{C}^1$. Recall that the boundary curve of $\mathcal{C}^1$ consists of a constant segment $q^1(p) \equiv s_{\min}^1$ on $J_{m^1}$, a strictly increasing part on the intervals $J_k$ for $1 < k < m^1$, and a vertical segment at $p = p_1^1 = s_{\max}^1$. Accordingly, if $\pi^0 \ge q^1(p_1^1)$ then $p_{\max}^1 = s_{\max}^1$; otherwise, $p_{\max}^1$ lies in the strictly increasing part and satisfies $q^1(p_{\max}^1) = \pi^0$.

To locate this value, consider the partition of the $q$-axis induced by $\{J_k\}$. Define $q_k := q(p_k)$, and since $p_k = p^*(\mu_k)$, we

---

**Algorithm 2** Compute $p_{\max}$ and $q_{\min}$.

---

**Input:** Scores and weights $(s_i^a, P(s_i^a | A = a))$ for each of the groups $a \in \{0, 1\}$, with scores in descending order
**Output:** $p_{\max}, q_{\min}$

$a \leftarrow \operatorname{argmax}_{a \in \{0,1\}} \pi^a$

$k \leftarrow \min\{k \in \{1, \ldots, m^a - 1\} \, : \, q_k^a \leq \pi^{1-a}\}$
**if** $k = 1$ **then**
$\quad p_{\max}^a \leftarrow s_{\max}^a$
**else**
$$p_{\max}^a \leftarrow \frac{(s_k^a + c_k^a)\pi^{1-a} - s_k^a \pi^a}{\pi^{1-a} - \pi^a + c_k^a}$$
**end if**
$p_{\max} \leftarrow \min\{p_{\max}^a, s_{\max}^{1-a}\}$

$k \leftarrow \min\{k \in \{2, \ldots, m^{1-a}\} \, : \, p_k^{1-a} \leq \pi^a\}$
**if** $k = m^{1-a}$ **then**
$\quad q_{\min}^{1-a} \leftarrow s_{\min}^{1-a}$
**else**
$$q_{\min}^{1-a} \leftarrow \frac{(\pi^{1-a} - c_k^{1-a})\pi^a - s_k^{1-a}\pi^{1-a}}{\pi^a - s_k^{1-a} - c_k^{1-a}}$$
**end if**
$q_{\min} \leftarrow \max\{q_{\min}^{1-a}, s_{\min}^a\}$

**return** $p_{\max}, q_{\min}$

---

obtain from (2)

$$
\begin{aligned}
q_k := q(p_k) &= \frac{\pi - \mu_k p_k}{1 - \mu_k} \\
&= \frac{\sum_{i \in [m]} s_i P(s_i) - \sum_{i \leq k} s_i P(s_i)}{1 - \sum_{i \leq k} P(s_i)} \\
&= \frac{\sum_{i > k} s_i P(s_i)}{\sum_{i > k} P(s_i)}.
\end{aligned}
$$

In particular, $q_{m^1-1}^1 = s_{m^1}^1 = s_{\min}^1$, so the intervals $J_k$ for $1 < k < m^1$ induce a partition of $q \in [s_{\min}^1, q^1(p_1^1))$.

Let $k^* := \min\{1 \leq k < m^1 \mid q_k^1 \leq \pi^0\}$. If $k^* = 1$, then $\pi^0 \geq q^1(p_1^1)$ and $p_{\max}^1 = s_{\max}^1$. Otherwise, $\pi^0 \in [q_k^1, q_{k-1}^1)$, hence $p_{\max}^1 \in J_k^1$. On this interval, the boundary is given by (7), so solving $q^1(p) = \pi^0$ yields

$$p_{\max}^1 = \frac{(s_{k^*}^1 + c_{k^*}^1)\pi^0 - s_{k^*}^1 \pi^1}{\pi^0 - \pi^1 + c_{k^*}^1}.$$

Note that the denominator is nonzero: if $\pi^0 - \pi^1 + c_k^1 = 0$, then the equation $q^1(p) = \pi^0$ forces $s_k^1 = \pi^0$ and hence $s_k^1 - \pi^1 + c_k^1 = 0$, contradicting (18) in Appendix C for $k < m^1$.

Finally, by symmetry, the boundary curve of $\mathcal{C}^0$ crosses (or attains) the value $p = \pi^1$ at $q_{\min}^0 := q^0(\pi^1)$, and all $(p, q) \in \mathcal{C}^0 \cap \mathcal{C}^1$ satisfy $q \geq \max\{q_{\min}^0, s_{\min}^1\}$. This specific value is not directly used by Algorithm 1, which parametrizes the boundary by $p$, but may be of interest when computing quantities (for example, a loss function) at the endpoints of the boundary curve of $\mathcal{C}^0 \cap \mathcal{C}^1$.

Let $k^* := \min\{1 < k \leq m^0 \mid p_k^0 \leq \pi^1\}$. If $k^* = m^0$, then $\pi^1 \in J_{m^0}^0$ and $q_{\min}^0 = s_{\min}^0$. Otherwise, $\pi^1 \in J_k$ for $1 < k < m^0$ and $q_{\min}^0$ is obtained by evaluating (7) at $p = \pi^1$.

The computation of $p_{\max}$ and $q_{\min}$ is summarized in Algorithm 2.

# F. Active Boundary Check

Let $p \in J_k^0 \cap J_l^1$ for some $1 < k \le m^0$ and $1 < l \le m^1$. Then by (7), $q^0(p) \ge q^1(p)$ iff

$$\frac{(\pi^0 - c_k^0)p - s_k^0 \pi^0}{p - s_k^0 - c_k^0} \ge \frac{(\pi^1 - c_l^1)p - s_l^1 \pi^1}{p - s_l^1 - c_l^1}.$$

Rearranging the terms, this condition is equivalent to

$$\Phi_{k,l}(p) = A_{k,l}p^2 + B_{k,l}p + C_{k,l} \ge 0,$$

where

$$
\begin{aligned}
A_{k,l} &:= (\pi^0 - c_k^0) - (\pi^1 - c_l^1), \\
B_{k,l} &:= s_l^1 \pi^1 - s_k^0 \pi^0 + (s_k^0 + c_k^0)(\pi^1 - c_l^1) - (s_l^1 + c_l^1)(\pi^0 - c_k^0), \\
C_{k,l} &:= (s_l^1 + c_l^1)s_k^0 \pi^0 - (s_k^0 + c_k^0)s_l^1 \pi^1.
\end{aligned}
$$

# G. Loss Minimization

Following the assumptions in Section 5, we are given a loss function $\ell : \{0,1\}^2 \to \mathbb{R}$ with $\ell_{00} = \ell_{11} = 0$ and $\ell_{01} + \ell_{10} > 0$. By (13), the expected loss of a classifier $R$ that attains a pair $(\pi, \pi) \ne (p, q) \in \mathcal{C}^0 \cap \mathcal{C}^1$ is

$$L(p,q) := \mathbb{E}[\ell(R, Y)] = \pi \ell_{01} + \frac{\pi - q}{p - q}(\ell_{10} - p(\ell_{01} + \ell_{10})),$$

where $\pi := P(A = 0)\pi^0 + P(A = 1)\pi^1$ is the aggregate base-rate of $Y$. Moreover, (12) implies that the minimum of $L(p,q)$ is achieved on the boundary of $\mathcal{C}^0 \cap \mathcal{C}^1$. In this appendix we show how to find the minimizer of $L(p,q)$ using Algorithm 1.

First, we prove a simple property of $L$ on line segments:

**Proposition G.1.** *Let $(p_1, q_1)$ and $(p_2, q_2)$ be the endpoints of a line segment contained in $\mathcal{C}^0 \cap \mathcal{C}^1$. If the segment is vertical $(p_1 = p_2 = p)$ or horizontal $(q_1 = q_2 = q)$, then the minimum of $L(p,q)$ on the segment is attained at one of the endpoints.*

*Proof.* Assume first that $p_1 = p_2 = p$. Differentiating (13) with respect to $q$ gives

$$
\begin{aligned}
\frac{dL}{dq}(q) &= (\ell_{10} - p(\ell_{01} + \ell_{10}))\frac{-(p-q) + (\pi - q)}{(p-q)^2} \\
&= (\ell_{10} - p(\ell_{01} + \ell_{10}))\frac{\pi - p}{(p-q)^2},
\end{aligned}
$$

whose sign is constant in $q$.

Conversely, if $q_1 = q_2 = q$, differentiating with respect to $p$ gives

$$
\begin{aligned}
\frac{dL}{dp}(p) &= (\pi - q)\frac{-(\ell_{01} + \ell_{10})(p - q) - (\ell_{10} - p(\ell_{01} + \ell_{10}))}{(p-q)^2} \\
&= (\pi - q)\frac{(\ell_{01} + \ell_{10})q - \ell_{10}}{(p-q)^2},
\end{aligned}
$$

whose sign is constant in $p$.

In either case, $L(p,q)$ is monotonic (or constant) along the segment and attains its minimum at an endpoint. $\square$

Let $(p,q) \in \partial(\mathcal{C}^0 \cap \mathcal{C}^1)$. As shown in Section 4.2, either $(p,q) \in J_{k,l,i}$ for some $k, l, i$, or it lies on a vertical segment at $p = p_{\max}$. Assume that $(p,q) \in J_{k,l,i}$ and, without loss of generality, assume that the active boundary on that interval is $\partial \mathcal{C}^0$ (otherwise, swap the roles of groups 0 and 1, and interchange $k$ and $l$). Then $q = q^0(p)$ is given by (7) and we obtain

$$L(p,q) = L(p) = \pi \ell_{01} + \frac{D_k^0 p + E_k^0}{(p - \pi^0)(p - s_k^0)}(\ell_{10} - p(\ell_{01} + \ell_{10})),$$

where

$$D_k^0 := \pi - \pi^0 + c_k^0,$$
$$E_k^0 := (\pi^0 - \pi)s_k^0 - c_k^0\pi$$

and $k$ is the respective index of $J_{k,l,i}$. Differentiating with respect to $p$ yields

$$\frac{dL}{dp}(p) = \frac{F_k^0 p^2 + G_k^0 p + H_k^0}{(p - \pi^0)^2(p - s_k^0)^2},$$

where

$$F_k^0 := (\ell_{01} + \ell_{10})\big(D_k^0(\pi^0 + s_k^0) + E_k^0\big) - \ell_{10}D_k^0,$$
$$G_k^0 := -2(\ell_{01} + \ell_{10})D_k^0\pi^0 s_k^0 - 2\ell_{10}E_k^0,$$
$$H_k^0 := \big(\ell_{10}D_k^0 - (\ell_{01} + \ell_{10})E_k^0\big)\pi^0 s_k^0 + \ell_{10}E_k^0(\pi^0 + s_k^0).$$

Therefore, the minimum of $L(p, q)$ on $J_{k,l,i}$ is attained at one of the endpoints or at a real root of $\Psi_k^0(p) := F_k^0 p^2 + G_k^0 p + H_k^0$ in the interior of $J_{k,l,i}$. Since Algorithm 1 iterates over all the intervals $J_{k,l,i}$, it suffices to evaluate $L(p, q)$ at the internal roots of $\Psi_k^0(p)$ and the right endpoint of each interval, as well as at the left endpoint of the leftmost interval, that is $(\max\{\pi^0, \pi^1\}, q_{\min})$ (see Appendix E for the definition of $q_{\min}$ and an algorithm to compute it).

Finally, if $p_{\max} = \min\{s_{\max}^0, s_{\max}^1\}$, the boundary of $\mathcal{C}^0 \cap \mathcal{C}^1$ contains a vertical segment at $p = p_{\max}$, and it is left to evaluate $L(p, q)$ there. By Proposition G.1, it suffices to evaluate $L(p, q)$ at the endpoints of the vertical segment, namely $(p_{\max}, q(p_{\max}^-))$ and $(p_{\max}, \pi^0)$. The lower endpoint coincides with the right endpoint of the rightmost interval $J_{k,l,i}$, hence it suffices to check $L(p_{\max}, \pi^0)$.

## H. Minimizing Deviation from Separation

The deviation from separation of a classifier $R$, using total variation divergence, is defined as (Benger & Ligett, 2025)

$$\Delta_{\text{sep}}(R) := \mathbb{E}_{A,Y} \, \text{TV}\big(P(R|A, Y), \, P(R|Y)\big)$$
$$= \sum_{a \in \{0,1\}} \sum_{y \in \{0,1\}} P(a)P(y|a) \, \text{TV}\big(P(R|a, y), \, P(R|y)\big),$$

where

$$\text{TV}\big(P(R|a, y), \, P(R|y)\big) := \frac{1}{2} \sum_{r \in \{0,1\}} |P(r|a, y) - P(r|y)|$$
$$= |P(R = 1|a, y) - P(R = 1|y)|.$$

Let $(\pi, \pi) \neq (p, q) \in \mathcal{C}^0 \cap \mathcal{C}^1$ be attained by $R$, then for all $a \in \{0, 1\}$, following Bayes' rule,

$$\text{TPR}(R) := P(R = 1|Y = 1) = \frac{P(R = 1)}{P(Y = 1)}P(Y = 1|R = 1) = \frac{\mu}{\pi}p,$$

$$\text{TPR}^a(R) := P(R = 1|Y = 1, a) = \frac{P(R = 1|a)}{P(Y = 1|a)}P(Y = 1|R = 1, a) = \frac{\mu^a}{\pi^a}p,$$

where we used the equality $p^a = p$, since $(p, q)$ belongs to the intersection of the feasible sets. Similarly,

$$\text{FPR}(R) := P(R = 1|Y = 0) = \frac{P(R = 1)}{P(Y = 0)}P(Y = 0|R = 1) = \frac{\mu}{1 - \pi}(1 - p),$$

$$\text{FPR}^a(R) := P(R = 1|Y = 0, a) = \frac{P(R = 1|a)}{P(Y = 0|a)}P(Y = 0|R = 1, a) = \frac{\mu^a}{1 - \pi^a}(1 - p).$$

Therefore,

$$\mathbb{E}_{Y|a} \, \mathrm{TV}\left(P(R|Y,a), P(R|Y)\right) = \pi^a p \left| \frac{\mu^a}{\pi^a} - \frac{\mu}{\pi} \right| + (1 - \pi^a)(1 - p) \left| \frac{\mu^a}{1 - \pi^a} - \frac{\mu}{1 - \pi} \right|. \tag{20}$$

By (2), $\mu = \frac{\pi - q}{p - q}$ and $\mu^a = \frac{\pi^a - q}{p - q}$, so

$$\mu^a = 1 - \frac{p - \pi^a}{p - q} = 1 - \frac{p - \pi^a}{p - \pi}(1 - \mu) = \frac{(\pi^a - \pi) + \mu(p - \pi^a)}{p - \pi}.$$

Substituting this expression inside the terms of (20) gives

$$\pi^a p \left| \frac{\mu^a}{\pi^a} - \frac{\mu}{\pi} \right| = \frac{p}{\pi(p - \pi)} \left| (\pi^a - \pi)(\pi - \mu p) \right|,$$

$$(1 - \pi^a)(1 - p) \left| \frac{\mu^a}{1 - \pi^a} - \frac{\mu}{1 - \pi} \right| = \frac{1 - p}{(1 - \pi)(p - \pi)} \left| (\pi^a - \pi)\left((1 - \pi) - \mu(1 - p)\right) \right|.$$

Since $\pi - \mu p = (1 - \mu)q \geq 0$ and $(1 - \pi) - \mu(1 - p) = (1 - \mu)(1 - q) \geq 0$, these terms can be taken out of the absolute values. Substituting into (20) and rearranging yields

$$\mathbb{E}_{Y|a} \, \mathrm{TV}\left(P(R|Y,a), P(R|Y)\right) = |\pi^a - \pi| \left( \frac{1 - \mu}{p - \pi} - \frac{\mu(p - \pi)}{\pi(1 - \pi)} \right).$$

Now, the only term that depends on $a$ is $|\pi^a - \pi|$ and we define

$$\begin{aligned}
K &:= \mathbb{E}_A \, |\pi^A - \pi| = P(A = 0)|\pi^0 - \pi| + P(A = 1)|\pi^1 - \pi| \\
&= P(A = 0)|\pi^0 - P(A = 0)\pi^0 - P(A = 1)\pi^1| + P(A = 1)|\pi^1 - P(A = 0)\pi^0 - P(A = 1)\pi^1| \\
&= P(A = 0)P(A = 1)|\pi^0 - \pi^1| + P(A = 1)P(A = 0)|\pi^1 - \pi^0| \\
&= 2P(A = 0)P(A = 1)|\pi^0 - \pi^1| \geq 0.
\end{aligned}$$

Therefore,

$$\Delta_{\mathrm{sep}}(R) = K \left( \frac{1 - \mu}{p - \pi} - \frac{\mu(p - \pi)}{\pi(1 - \pi)} \right). \tag{21}$$

As shown in Section 6, differentiating with respect to $p$ (holding $\mu$ fixed) we obtain

$$\frac{d}{dp} \Delta_{\mathrm{sep}}(R) = K \left( -\frac{1 - \mu}{(p - \pi)^2} - \frac{\mu}{\pi(1 - \pi)} \right) \leq 0,$$

so for a fixed $\mu$ the deviation from separation is constant or minimized by maximizing $p$, implying that a minimizer is attained on the boundary of $\mathcal{C}^0 \cap \mathcal{C}^1$.

Finally, we substitute for $\mu = \frac{\pi - q}{p - q}$ in (21), so

$$\Delta_{\mathrm{sep}}(R) = \Delta_{\mathrm{sep}}(p, q) = \tilde{K} \frac{\pi(1 - p) + q(p - \pi)}{p - q}, \tag{22}$$

where

$$\tilde{K} := \frac{K}{\pi(1 - \pi)} = \frac{2P(A = 0)P(A = 1)|\pi^0 - \pi^1|}{\pi(1 - \pi)}.$$

We now show how the minimizer of $\Delta_{\mathrm{sep}}(R)$ can be found using Algorithm 1, following the same steps of Appendix G for attaining the loss minimizer.

First, similar to the case with expected loss (see Proposition G.1), $\Delta_{\mathrm{sep}}(R)$ also behaves nicely on line segments:

**Proposition H.1.** *Let $(p_1, q_1)$ and $(p_2, q_2)$ be the endpoints of a line segment contained in $\mathcal{C}^0 \cap \mathcal{C}^1$. If the segment is vertical ($p_1 = p_2 = p$) or horizontal ($q_1 = q_2 = q$), then the minimum of $\Delta_{\mathrm{sep}}(p, q)$ on the segment is attained at one of the endpoints.*

*Proof.* Assume first that $p_1 = p_2 = p$. Differentiating (22) with respect to $q$ gives

$$\frac{d\Delta_{\text{sep}}}{dq}(q) = \tilde{K}\frac{p^2 - 2\pi p + \pi}{(p - q)^2}$$

whose sign is constant in $q$.

Conversely, if $q_1 = q_2 = q$, differentiating with respect to $p$ gives

$$\frac{d\Delta_{\text{sep}}}{dp}(p) = -\tilde{K}\frac{q^2 - 2\pi q + \pi}{(p - q)^2}$$

whose sign is constant in $p$.

In either case, $\Delta_{\text{sep}}(p, q)$ is monotonic (or constant) along the segment and attains its minimum at an endpoint. $\qquad\square$

Let $(p, q) \in \partial(\mathcal{C}^0 \cap \mathcal{C}^1)$. Assume that $(p, q) \in J_{k,l,i}$ and, without loss of generality, assume that the active boundary on that interval is $\partial\mathcal{C}^0$. Substituting $q = q^0(p)$ in (22) we obtain

$$\Delta_{\text{sep}}(p, q) = \Delta_{\text{sep}}(p) = \tilde{K}\frac{\tilde{A}_k^0 p^2 + \tilde{B}_k^0 p + \tilde{C}_k^0}{(p - \pi^0)(p - s_k^0)},$$

where

$$\tilde{A}_k^0 := \pi^0 - \pi - c_k^0,$$
$$\tilde{B}_k^0 := \pi(2c_k^0 - \pi^0 + s_k^0 + 1) - s_k^0\pi^0,$$
$$\tilde{C}_k^0 := \pi(s_k^0\pi^0 - s_k^0 - c_k^0).$$

Differentiating with respect to $p$ yields

$$\frac{d\Delta_{\text{sep}}}{dp}(p) = \tilde{K}\frac{\tilde{D}_k^0 p^2 + \tilde{E}_k^0 p + \tilde{F}_k^0}{(p - \pi^0)^2(p - s_k^0)^2},$$

where

$$\tilde{D}_k^0 := -\tilde{A}_k^0(\pi^0 + s_k^0) - \tilde{B}_k^0,$$
$$\tilde{E}_k^0 := 2\tilde{A}_k^0\pi^0 s_k^0 - 2\tilde{C}_k^0,$$
$$\tilde{F}_k^0 := \tilde{B}_k^0\pi^0 s_k^0 + \tilde{C}_k^0(\pi^0 + s_k^0).$$

The minimum of $\Delta_{\text{sep}}(p, q)$ on $J_{k,l,i}$ is attained at one of the endpoints or at a real root of $\tilde{\Psi}_k^0(p) := \tilde{D}_k^0 p^2 + \tilde{E}_k^0 p + \tilde{F}_k^0$ in the interior of $J_{k,l,i}$. Since Algorithm 1 iterates over all the intervals $J_{k,l,i}$, it suffices to evaluate $\Delta_{\text{sep}}(p, q)$ at the internal roots of $\tilde{\Psi}_k^0(p)$ and the right endpoint of each interval, as well as at the left endpoint of the leftmost interval, that is $(\max\{\pi^0, \pi^1\}, q_{\min})$ (see Appendix E for the definition of $q_{\min}$ and an algorithm to compute it).

Finally, if $p_{\max} = \min\{s_{\max}^0, s_{\max}^1\}$, the boundary of $\mathcal{C}^0 \cap \mathcal{C}^1$ contains a vertical segment at $p = p_{\max}$, and it is left to evaluate $\Delta_{\text{sep}}(p, q)$ there. By Proposition H.1, it suffices to evaluate $\Delta_{\text{sep}}(p, q)$ at the endpoints of the vertical segment, namely $(p_{\max}, q(p_{\max}^-))$ and $(p_{\max}, \pi^0)$. The lower endpoint coincides with the right endpoint of the rightmost interval $J_{k,l,i}$, hence it suffices to check $\Delta_{\text{sep}}(p_{\max}, \pi^0)$.

## I. Experiment Details

### I.1. FICO

We use the public FICO/TransRisk aggregate statistics accompanying Barocas et al. (2023).[6] The data contain, for each race or ethnicity, the total number of individuals, the cumulative distribution of credit-score bins, and the empirical performance

---

[6] https://github.com/fairmlbook/fairmlbook.github.io/tree/master/code/creditscore/data

associated with each bin. We restrict attention to the groups labeled "Non-Hispanic white" and "Black" in the data, referred to below as "white" and "black", and denoted $A = 0$ and $A = 1$, respectively.

The reported performance value is the percentage of "bad" outcomes in each TransRisk bin. Since we take $Y = 1$ to denote non-default, we convert it to a non-default probability by setting

$$s_i^a := 1 - \text{perf}_i^a / 100.$$

The group-specific bin weights are obtained by differencing the empirical CDFs,

$$P(s_i^a \mid A = a) = \left(F_i^a - F_{i-1}^a\right)/100,$$

and the group proportion is

$$P(A = 1) = \frac{n_{\text{black}}}{n_{\text{white}} + n_{\text{black}}}.$$

These quantities give the inputs $(s_i^a, P(s_i^a | A = a))$ and $P(A = a)$ for Algorithm 1. Since the data are already given as aggregate score statistics, no train-test split or model-fitting step is involved.

As a baseline, we compute the unconstrained group-aware Bayes classifier, which thresholds the calibrated score at $\frac{1}{2}$ separately within each group. This classifier attains accuracy 0.8819, but violates predictive parity: its PPV values are 0.9101 and 0.7854, and its FOR values are 0.1980 and 0.1289, for the "white" and "black" groups, respectively.

Applying Algorithm 1 with the 0–1 loss objective yields the accuracy-optimal sufficient classifier, attaining

$$(p, q) = (0.9116, 0.2346)$$

and accuracy 0.8676. The optimum lies on $\partial\mathcal{C}^1$ but in the interior of $\mathcal{C}^0$; hence, the selection rule for the "black" group is a threshold, while the selection rule for the "white" group cannot be realized by thresholding alone. One possible realization of the "white" group rule is obtained by the mixture construction following Theorem 3.3.

For comparison, the classifier minimizing deviation from separation attains

$$(p, q) = (0.9284, 0.2607),$$

with $\Delta_{\text{sep}} = 0.0702$ and accuracy 0.8659. Thus, in this example, minimizing deviation from separation yields accuracy very close to that of the accuracy-optimal sufficient classifier.

### I.2. COMPAS

We use the ProPublica COMPAS two-year recidivism data (Angwin et al., 2016).[7] We apply the standard filters used in the ProPublica analysis: we keep only individuals whose COMPAS screening date is within 30 days of arrest, remove observations with invalid recidivism labels, ordinary traffic offenses, or missing score labels, and restrict attention to the groups labeled "Caucasian" and "African-American" in the data, referred to below as "white" and "black", and denoted $A = 0$ and $A = 1$, respectively. We take $Y = 1$ to indicate recidivism within two years, and let $d \in \{1, \dots, 10\}$ denote the COMPAS decile score.

Unlike in the FICO experiment, official aggregate score statistics are not available. Motivated by the approximate sufficiency of COMPAS scores (Chouldechova, 2017), we construct group-calibrated aggregate inputs as follows. For each decile score $d$, we assign the pooled empirical outcome rate

$$s_d := \frac{\sum_i \mathbf{1}\{d_i = d\}Y_i}{\sum_i \mathbf{1}\{d_i = d\}},$$

ignoring group membership. The group-specific weights of these score values are then estimated by

$$P(s_d \mid A = a) = \frac{\sum_i \mathbf{1}\{A_i = a, d_i = d\}}{\sum_i \mathbf{1}\{A_i = a\}},$$

---

[7] https://github.com/propublica/compas-analysis

and the group proportion is

$$P(A = 1) = \frac{\sum_i \mathbf{1}\{A_i = 1\}}{n}.$$

Thus, group membership affects the feasible regions only through the group-specific distributions of decile scores. No classifier is trained and no train-test split is used in this experiment; the purpose is to illustrate our post-processing method directly on the COMPAS decile score.

As a baseline, we compute the unconstrained Bayes classifier under the constructed score model, obtained by thresholding the pooled empirical score $s_d$ at $\frac{1}{2}$. This classifier attains accuracy 0.6629, but violates predictive parity: its PPV values are 0.6611 and 0.6807, and its FOR values are 0.3273 and 0.3620, for the "white" and "black" groups, respectively.

Applying Algorithm 1 with the 0–1 loss objective yields the accuracy-optimal sufficient classifier, attaining

$$(p, q) = (0.6446, 0.3346)$$

and accuracy 0.6563. The optimum lies on a breakpoint of $\partial \mathcal{C}^1$, corresponding to the deterministic threshold that selects all individuals with decile score at least 5 in the "black" group. In contrast, the corresponding point for the "white" group lies in the interior of $\mathcal{C}^0$, and therefore cannot be realized by thresholding alone. The nearest breakpoint of $\partial \mathcal{C}^0$ corresponds to the deterministic threshold selecting all individuals from the "white" group with decile score at least 6.

For comparison, the classifier minimizing deviation from separation attains

$$(p, q) = (0.6807, 0.3620),$$

with $\Delta_{\text{sep}} = 0.1464$ and accuracy 0.6526, again close to the accuracy-optimal sufficient classifier.

### I.3. ACS Income

We use the ACS Income dataset (Ding et al., 2021), downloaded from OpenML,[8] and restrict attention to the California subset. The task is to predict whether an individual's annual income exceeds \$50K, using the ACS personal income variable. We use sex as the protected attribute, with $A = 0$ for the "male" group and $A = 1$ for the "female" group. The prediction model uses age, class of worker, educational attainment, marital status, occupation, place of birth, relationship to householder, hours worked per week, sex, and race. Numerical features are standardized and categorical features are one-hot encoded, grouping infrequent categories.

We repeat the experiment over 100 random train/test splits, each with a held-out test set containing 20% of the data and stratified by $(A, Y)$. In each repetition, we train a logistic regression model using stochastic gradient descent with log-loss and $\ell_2$ regularization. To obtain calibrated scores, we compute out-of-fold predicted probabilities on the training set using 5-fold stratified cross-validation. For each group separately, we divide these out-of-fold predictions into $k = 100$ equal-mass bins and assign each bin the empirical mean of $Y$ in that bin. We then fit the model on the full training set, compute predicted probabilities on the test set, and assign each test point to the corresponding calibration-derived bin.

The inputs to Algorithm 1 are constructed as follows. The score values $s_i^a$ are the empirical label means of the calibration bins described above. The bin weights $P(s_i^a \mid A = a)$ are estimated from the held-out test set, using the fraction of test points in group $a$ assigned to each calibration-derived bin. Thus, the binning and calibrated score values are estimated from the training data, while the score-bin weights are estimated from the held-out test scores and group membership, without using test labels. We evaluate the resulting randomized classifier in expectation, using its selection probabilities. As an unconstrained baseline, we threshold the raw logistic-regression output at $\frac{1}{2}$.

Over the 100 repetitions, the post-processed classifier attains average group-wise values $\text{PPV}^0 = 0.7758$, $\text{PPV}^1 = 0.7768$, $\text{FOR}^0 = 0.1545$, and $\text{FOR}^1 = 0.1549$. The mean absolute PPV gap is $4.8 \times 10^{-3}$ $[4.2 \times 10^{-3}, 5.5 \times 10^{-3}]$, and the mean absolute FOR gap is $3.7 \times 10^{-3}$ $[3.2 \times 10^{-3}, 4.3 \times 10^{-3}]$ (brackets denote 95% percentile bootstrap confidence intervals for the mean over repetitions, computed using 10,000 resamples). The mean accuracy is 0.8167 $[0.8164, 0.8171]$.

For comparison, the unconstrained classifier obtains $\text{PPV}^0 = 0.7961$, $\text{PPV}^1 = 0.7600$, $\text{FOR}^0 = 0.1728$, and $\text{FOR}^1 = 0.1400$. Its mean absolute PPV gap is $3.61 \times 10^{-2}$ $[3.51 \times 10^{-2}, 3.71 \times 10^{-2}]$, and its mean absolute FOR gap is $3.29 \times 10^{-2}$ $[3.21 \times 10^{-2}, 3.36 \times 10^{-2}]$. The mean accuracy of the unconstrained classifier is 0.8190 $[0.8187, 0.8194]$.

Figure 5 shows the distribution of these quantities over the 100 repetitions.

---

[8] We use the OpenML version accessed via `sklearn.datasets.fetch_openml` with `data_id=43141`.

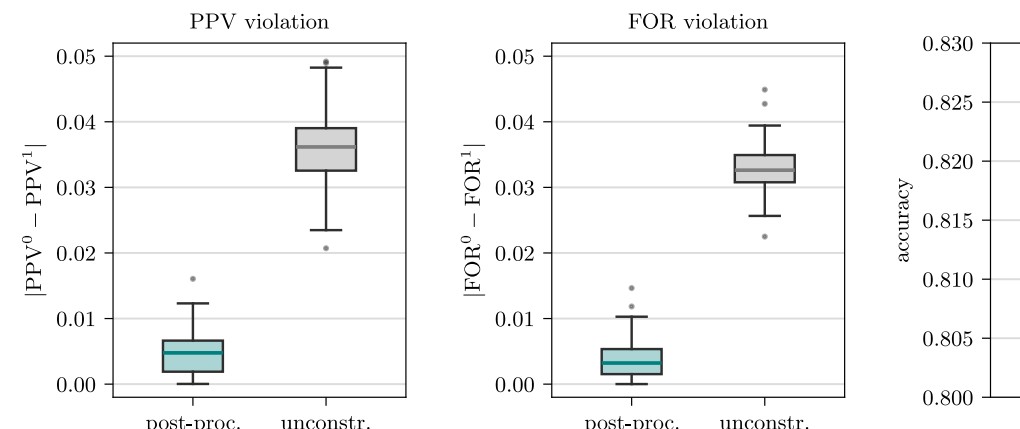

*Figure 5.* **ACS Income.** Results over 100 train/test splits. The post-processed classifier is obtained by Algorithm 1; the unconstrained baseline thresholds the raw logistic-regression output at $\frac{1}{2}$. The post-processed classifier substantially reduces the PPV and FOR gaps, with only a small loss in accuracy. Boxes denote interquartile ranges, center lines denote medians, whiskers extend to 1.5 times the interquartile range, and dots denote outliers.

## J. Robustness to Calibration Error

The assumption that $S$ is group-calibrated is central to this work. However, in practice, one might rely on predictions or scores that are only approximately group-calibrated. In this appendix, we show that the predictive parity violation that may result from using our method with approximately calibrated scores is bounded linearly by the maximum calibration error. We illustrate empirically that the violations in practice are considerably lower than this bound, and that the achieved accuracy is stable under such perturbations.

Ignoring the role of $A$, a classifier $R$ with selection rule $P(R|s_i)$ attains

$$\text{PPV}(R) = \frac{1}{\mu} \sum\nolimits_{i \in [m]} P(Y = 1|s_i) \, P(s_i) \, P(R = 1|s_i),$$

where $\mu = P(R = 1)$. If the scores $S$ are calibrated, this value equals the score-based quantity

$$p = \frac{1}{\mu} \sum\nolimits_{i \in [m]} s_i \, P(s_i) \, P(R = 1|s_i) \tag{23}$$

(see Equation (4)). Denoting the (maximum) calibration error of $S$ by $\epsilon := \max_{i \in [m]} |s_i - P(Y = 1|s_i)|$, we get

$$|p - \text{PPV}(R)| \leq \frac{1}{\mu} \sum\nolimits_{i \in [m]} |s_i - P(Y = 1|s_i)| \, P(s_i) \, P(R = 1|s_i)$$
$$\leq \left( \frac{1}{\mu} \sum\nolimits_{i \in [m]} P(s_i) \, P(R = 1|s_i) \right) \max_{i \in [m]} |s_i - P(Y = 1|s_i)| = \epsilon,$$

where the last equality follows from (3).

Now, suppose that $R$ attains predictive parity with respect to $A$, under the assumption that the scores $S$ are perfectly group-calibrated. This means that $p^0 = p^1 = p$. If the scores are not perfectly group-calibrated, denote by $\epsilon := \max_{a \in \{0,1\}} \max_{i \in [m^a]} |s_i^a - P(Y = 1|s_i^a, a)|$ the global calibration error. Then, by the bound above,

$$|\text{PPV}^0(R) - \text{PPV}^1(R)| \leq |\text{PPV}^0(R) - p^0| + |p^0 - p^1| + |p^1 - \text{PPV}^1(R)|$$
$$= |\text{PPV}^0(R) - p^0| + |p^1 - \text{PPV}^1(R)| \leq 2\epsilon.$$

Analogously, $|\text{FOR}^0(R) - \text{FOR}^1(R)| \leq 2\epsilon$.

In other words, if our algorithm is run using miscalibrated scores, the resulting predictive parity violations, measured by the absolute group-wise differences in PPV and FOR, are each at most twice the maximum calibration error.

### J.1. Synthetic Experiment

To illustrate this result, we use the same setting as in the left pane of Figure 2: $s^0 = (0.1, 0.2, 0.5, 0.7, 0.9)$, $P(s^0|A = 0) = (0.1, 0.3, 0.3, 0.15, 0.15)$; $s^1 = (0.12, 0.3, 0.85)$, $P(s^1|A = 1) = (0.15, 0.45, 0.4)$; and $P(A = 1) = \frac{1}{2}$. The optimal fair classifier (maximizing accuracy) attains $p = 0.80$, $q = 0.31$, with accuracy $0.7239$.

For ten values of $0.01 \leq \epsilon \leq 0.10$, we add random uniform perturbations to the scores such that $\max_{a \in \{0,1\}} \max_{i \in [m^a]} |s_i^a - \tilde{s}_i^a| = \epsilon$. The optimal selection rule is computed using Algorithm 1 on the perturbed scores $\tilde{s}$, and then evaluated on the original calibrated scores $s$. This is repeated 100 times for each value of $\epsilon$. The results, shown in Figure 6, are consistent with the $2\epsilon$ bound on the violation of predictive parity, while the observed differences are typically considerably below that bound. In addition, the achieved accuracy remains close to the true optimal value.

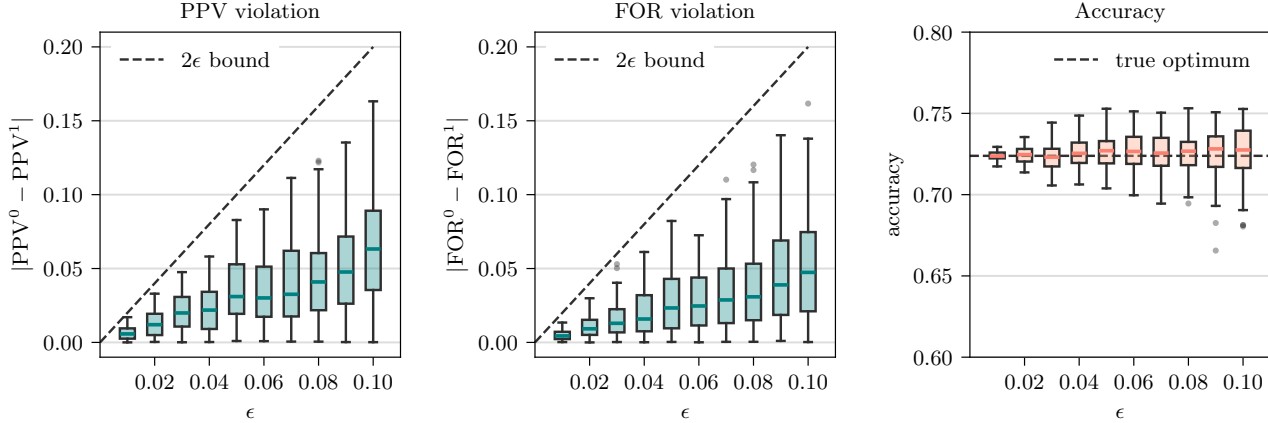

*Figure 6.* **Synthetic calibration-error experiment.** For each value of $\epsilon$, boxplots show the results over 100 random uniform perturbations of the score values, rescaled so that the maximum perturbation equals $\epsilon$. The dashed lines in the first two panels show the theoretical $2\epsilon$ bound on the PPV and FOR violations; the dashed line in the right panel shows the optimal accuracy under the original calibrated scores. Boxes denote interquartile ranges, center lines denote medians, whiskers extend to 1.5 times the interquartile range, and dots denote outliers.

### J.2. ACS Income

To illustrate the effect of approximate calibration on real-world data, we use the same setting as in the ACS Income experiment (see Appendix I.3). To induce different levels of calibration error, we subsample fractions $\rho$ of the calibration set and use only the resulting subset to estimate the group-wise score bins. Specifically, for each calibration subset, we recompute the group-wise bin edges and bin values, and then use these edges to bin the fixed test-set scores; the bin weights are computed from the resulting test-bin masses. The results, shown in Figure 7, exhibit a clear pattern: smaller calibration fractions tend to lead to larger predictive parity violations. Nevertheless, when using at least $10\%$ of the original calibration set, the absolute differences between the group-wise PPV and FOR remain typically below $1\%$, and the attained accuracy is less than one percentage point below the average accuracy of the unconstrained classifier.

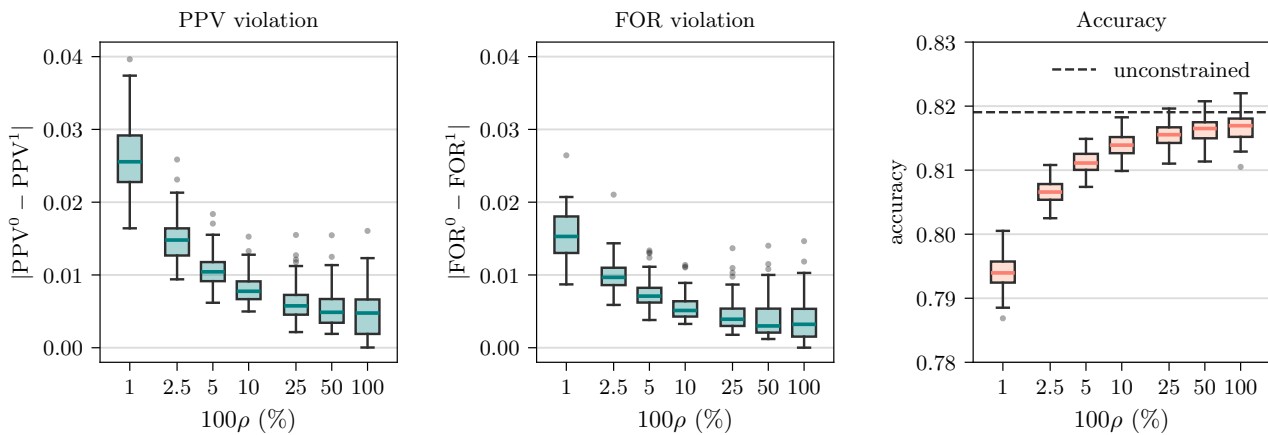

*Figure 7.* **ACS Income calibration-subsampling experiment.** We fix the number of calibration bins at 100, as in the original experiment, and vary only the fraction $\rho$ of the calibration data used to estimate the group-wise binning. For each train/test split and calibration fraction $\rho < 1$, results are averaged over 20 calibration subsamples; for $\rho = 1$, the full calibration set is used once. Boxplots show variation over 100 train/test splits. The horizontal axis is logarithmically spaced, and tick labels show the percentage $100\rho$ of the full calibration set used for calibration. The dashed line in the accuracy panel shows the mean unconstrained accuracy.

