# OpenReview forum: "Fair Decisions from Calibrated Scores: Achieving Optimal Classification While Satisfying Sufficiency"
_ICML.cc/2026/Conference — ICML 2026 regular_

### Official Review · Reviewer_Zqua · 2026-03-04

**Soundness:** 3
**Presentation:** 2
**Significance:** 3
**Originality:** 4
**Overall Recommendation:** 4
**Confidence:** 4

**Summary:**

This paper studies post-processing of group-calibrated scores to produce binary decisions that satisfy sufficiency across sensitive groups while remaining optimal for a given objective. The key technical contribution is a geometric characterization of the set of achievable  positive-predictive-value (PPV) and false-omission-rate (FOR) pairs for a single group under randomized score-based decision rules, and the observation that sufficiency across groups reduces to finding a common PPV-FOR pair in the intersection of group-wise feasible regions. Building on this, the authors propose an algorithm to trace the “active boundary” of the intersection and optimize the objective over that boundary, yielding a threshold rule with possible within-bin randomization.

**Compliance With Llm Reviewing Policy:**

Affirmed.

**Final Justification:**

My final recommendation is Weak Accept. I think the paper addresses an interesting and relevant problem, and the proposed method has solid technical merit. The main strengths are the soundness of the approach and the potential usefulness of the contribution. At the same time, I believe there are still some weaknesses that prevent me from rating it more strongly, particularly in terms of overall empirical support and the strong assumptions of perfectly calibrated scores.

The authors’ rebuttal addressed my main questions and resolved the concerns I raised in my initial review. Overall, I believe the paper makes a worthwhile contribution, and I therefore support acceptance.

**Key Questions For Authors:**

Q1.  What is the precise definition of $\mathcal{C}_{P(Y,S)} $ ? I could not find a formal definition in the main text., along with its connection to $(p,q) $ .

Q2. I found the beginning of Section 3 difficult to follow. Could you clarify the motivation on Equation (1), and explain the role of the argument around lines 113–116 at this point in the part? Presenting these steps so early (before the feasible-region intuition is established) makes it hard to see why they are introduced and how they will be used later.

Q3. The derived optimal decision rule has a form reminiscent of a Neyman–Pearson–type classifier (e.g., thresholding a likelihood-ratio–like quantity with possible randomization). Could the authors discuss more explicitly how this rule relates to prior post-processing methods for calibrated scores and/or classical optimality results under constraints, and clarify what is new relative to existing approaches?

**Limitations:**

A major limitation of the framework is its requirement on exact group calibration, whereas real-world scoring pipelines often produce approximately calibrated scores.

**Strengths And Weaknesses:**

**Strength**: This paper proposes an interesting and novel approach for constructing decision rules that satisfy sufficiency while optimizing a target objective. The geometric viewpoint, characterizing the set of achievable (PPV,FOR) pairs and then optimizing over the boundary of the feasible intersection, offers a clean way to handle sufficiency constraints and leads to implementable search algorithms. Overall, the approach is practically motivated and theoretically well grounded under the stated assumptions.


**Weakness 1**: The exposition would benefit from more intuition and a clearer “big picture” early on. In particular, Section 3 introduces the feasible region construction, but it is not immediately explained why this object is central, nor how it will be used to enforce sufficiency across groups; this connection only becomes clear in Section 4. As a result, readers may find Section 3 hard to place conceptually.

Also, it would help to include an explicit high-level optimization formulation near the beginning. My understanding (may be inaccurate, just a suggestion) is that the overall goal is to find group-dependent decision rules $R $ such that the induced predictive values are equalized across groups:
$$\arg\min_R Obj(R)\quad s.t. p^a(R)=p(R), q^a(R)=q(R) \; a\in Supp A;  $$

Using the feasible-region characterization, this can be reduced to selecting suitable $\mathbb{p}=(p^a)_{a\in Supp A} $  as

$$\arg\min_{\mathbb{p}} Obj(\mathbb{p})\quad s.t. C(\mathbb{p})=0, $$

where $C $ is a constraint on $\mathbb{p} $ such that $(p^a,q^a) $ are all at the intersection of feasible regions. Once $p^a $ is chosen, one can construct per-group rules $R $ that realize it. Stating similar roadmap up front would help to improve interpretability.


**Weakness 2**: The experimental evaluation appears limited. Only one real-data example is considered and there are no comparisons to existing baselines. The paper would be more convincing with a broader empirical study, including additional datasets or synthetic simulations as well as comparisons against stronger post-processing baselines for sufficiency.


**Weakness 3**: The theory assumes perfect group calibration and knowledge of the score distribution $\Pr(S=s\mid A=a) $ . In practice, calibration is approximate and both calibration and $\Pr(S=s\mid A=a) $ must be estimated from data. The paper would benefit from discussing robustness to calibration error and distribution estimation error—either theoretically or empirically.

---

> ### Author Rebuttal · Authors · 2026-03-31
>
> Thanks for your feedback!
>
> Regarding the comments:
>
> **W1** Thank you, we will improve the exposition with emphasis on the "bigger picture", so the structure of Sections 3 and 4 - from a single group to the fairness question - becomes clearer. The objectives you state are indeed the goal in Sections 5 and 6. The previous sections provide the geometric characterization that culminates in Algorithm 1, which can then be used to achieve those goals.
>
> **W2** Thank you for this suggestion. We added two experiments to broaden the empirical evaluation (see below).
>
> Regarding comparisons with existing baselines, there are relatively few methods that construct binary classifiers satisfying sufficiency, and most consider relaxed or approximate notions of sufficiency. Our method, in contrast, enforces exact equality of both PPV and FOR and optimizes the objective within this constraint. There exist works that achieve exact sufficiency, such as Canetti et al. (2019) and Baumann et al. (2022), but they rely on different modeling assumptions (e.g., allowing abstention or assuming continuous scores with full support), and are thus not directly comparable to our setting.
>
> We will clarify this distinction and better position our contribution in the paper.
>
> **W3** We note that our assumption of perfect calibration is weaker than the one typically made in the literature (e.g., Hardt, Price & Srebro (2016); Canetti et al. (2019); Baumann et al. (2022)), which assumes knowledge of the true probabilities $P(Y=1|X)$. Yet, we agree that it is interesting to explore the impact of using approximately group-calibrated scores. Here is a sketch of a simple analysis and two additional experiments, which we could incorporate. They suggest that the impact of imperfectly group-calibrated scores is small and approximately linear in the distance from group-calibration.
>
> Assuming the scores $s$ are estimated with error $\epsilon$, our analysis implies an error of $O(\epsilon)$ in the estimate of $p$ for any fixed $\mu$ (see, e.g., Eq. 6), hence also in the estimated boundary (if the perturbation is not large enough to "flip" the order of the scores). Eq. 12 similarly implies an $O(\epsilon)$ error in the expected loss.
>
> Additional experiments:
>
> **(a)** For the same **synthetic example** as in Fig. 2, we perturbed the scores by adding increasing random noise and then computed the optimal selection rules using Alg. 1 (minimizing 0-1 loss). As expected, these rules achieve perfect predictive parity on the perturbed scores. When evaluated on the original ("true") scores, the average differences in PPV and FOR between the groups do not exceed a few percentage points and follow an approximately linear function in the noise level.
>
> **(b)** We trained a logistic regressor on the **Adult dataset** with “sex” as the protected attribute, and calibrated its scores (by group) with isotonic regression. We induced varying levels of calibration error by varying the size of the calibration subset, then evaluated the predictors on a held-out test set using Alg. 1 to compute the optimal classifier. Even if only half of the training set is used to estimate the calibrated scores, the average difference in PPV between the groups is below 3%, and even smaller in FOR. Again, the violation follows an approximately linear relationship with calibration error.
>
> Figures of both experiments are in https://drive.google.com/file/d/1ixDjtNBDej-AlI3B21N6aJQgxcdFY7xp/view
>
> With respect to the questions:
>
> **Q1** The definition of $\mathcal C$ is given in Def. 3.1 (line 124). In mathematical notation:
> $\mathcal C_{P(Y,S)} := \\{ (p, q)\in[0,1]^2 \mid Y \leftrightarrow S \leftrightarrow R \\ \text{is Markov,} \\ PPV(R)=p, FOR(R)=q \\}$
>
> **Q2** Eq. 1 expresses a property of calibrated scores (their expected value equals that of $Y$), which we use in our analysis. The example in lines 113-116 gives intuition for the feasible set: calibration and Markovity together impose constraints on possible values of PPV and FOR - not everything is feasible. As an example, we show that PPV is bounded by the maximal score.
>
> **Q3** While the resulting decision rule may resemble thresholding in some cases, the structure differs from Neyman-Pearson-type solutions. In our setting, the constraint is not in likelihood ratios but in predictive values (PPV, FOR), depending nonlinearly on the decision rule. As a result, the optimal rule is characterized via the geometry of the feasible region and may correspond to an interior point of one group's feasible set, requiring within-bin randomization.
>
> Prior works (e.g., Canetti et al. (2019), Baumann et al. (2022)) also observe that thresholding alone is unable to achieve sufficiency, but do not provide a general method for constructing loss-optimal binary classifiers under exact sufficiency with discrete scores. Our contribution is precisely this geometric characterization and the resulting algorithm.
>
> We addressed the limitation in the answer to W3, above.

---

> > ### Author Rebuttal · Reviewer_Zqua · 2026-04-01
> >
> > Thanks to the authors for their response. I will keep my score.

---

### Official Review · Reviewer_3gdC · 2026-03-09

**Soundness:** 4
**Presentation:** 3
**Significance:** 3
**Originality:** 3
**Overall Recommendation:** 5
**Confidence:** 2

**Summary:**

This paper studies optimal binary classification under sufficiency when scores are finite-valued and group-calibrated. The key observation is that, unlike independence or separation, sufficiency is not preserved by thresholding; even perfectly calibrated scores violate predictive parity after a single threshold is applied. The authors characterize the feasible (PPV, FOR) region as a star-convex set with a piecewise-hyperbolic boundary, connect the optimal selection rule to a fractional knapsack problem, and propose a boundary-tracing algorithm to find the optimal sufficient classifier under both loss minimization and deviation-from-separation objectives. The method is demonstrated on the COMPAS dataset, where the fairness cost in accuracy is minimal.

**Compliance With Llm Reviewing Policy:**

Affirmed.

**Final Justification:**

I keep the positive score and low confidence.

**Key Questions For Authors:**

1. Can the loss-optimal sufficient classifier ever lie on a degenerate edge when the non-trivial intersection is nonempty?
2. How sensitive is the method to approximate (rather than exact) group calibration? Can the fairness violation or suboptimality be bounded as a function of calibration error?
3. How does Algorithm 1 generalize to g > 2 groups, and how does its complexity scale?

**Limitations:**

Several limitations could be further discussed:
1. No analysis of how finite-sample calibration error propagates to the optimality and fairness guarantees.
2. No complexity analysis for g > 2 groups.

**Strengths And Weaknesses:**

Strength:
1. The feasible-region characterization and its knapsack connection (Theorem 3.3) are clean, giving both a constructive selection rule and a clear optimality proof.
2. This paper handles finite-valued scores exactly, matching many real-world scoring systems.
3. Both loss minimization and separation-deviation minimization reduce to piecewise quadratic optimization along the same boundary, allowing direct comparison within one framework.

Weakness:
1. Only one dataset (COMPAS) with one binary protected attribute is used. It is hard to judge how the method behaves with more score bins, noisier calibration, or different group structures.
2. The entire framework assumes exact group calibration. Sensitivity to calibration error, inevitable with finite data, is not discussed.

---

> ### Author Rebuttal · Authors · 2026-03-31
>
> Thanks for your feedback!
>
> Addressing your comments:
>
> **1.**
>
> **(a)** We added an experiment on the Adult dataset. Specifically, we use it to explore the sensitivity to calibration error (see details below).
>
> **(b)** Regarding the number of score bins: as the number of bins increases, the boundary of the feasible sets becomes smoother (i.e., composed of smaller arcs), potentially allowing lower loss due to higher resolution. An illustration of this effect for the Adult dataset, with varying numbers of score bins, is provided in https://drive.google.com/file/d/1nDHkR3VHPIhAKpFEp0fqWu2_WDnzWAGH/view. We note that the number of bins is not a model parameter, but is determined by the scoring system in a given application (e.g., COMPAS provides 10 scores, while the SAT scoring scale has 121 possible values).
>
> **(c)** Regarding group structures and binary protected attributes: for more than two groups, there is no guarantee that a common intersection exists; this reflects the fact that a binary classifier satisfying sufficiency for all groups may not exist. Importantly, this is a structural property of the given score distributions, and it depends on the relative geometry of the group-wise feasible sets. As the number of groups increases, the intersection is taken over more sets and therefore becomes more restrictive, so infeasibility may arise more often in practice.
>
> From a computational perspective, if a nonempty intersection exists, the extension of our algorithm is straightforward: the boundary is determined by the pointwise maximum over all group-wise boundary functions (see Eq. 10), and the same piecewise analysis applies. The complexity scales linearly in the number of groups (up to the number of boundary segments considered), without introducing new conceptual difficulties.
>
> **2.**
>
> We note that our assumption of perfect calibration is actually weaker than the one typically made in the literature (e.g., Hardt, Price & Srebro (2016); Canetti et al. (2019); Baumann et al. (2022)), which assumes knowledge of the true probabilities $P(Y=1|X)$. Nonetheless, we agree that it is interesting to explore the impact of using approximately group-calibrated scores. Here is the sketch of a  simple analysis and two additional experiments, versions of which we could incorporate into the final version of the paper. These suggest that the impact of imperfectly group-calibrated scores is expected to be small and approximately linear in the distance from group-calibration.
>
> Assuming the scores $s$ are estimated with error $\epsilon$, our theoretical analysis implies an error of $O(\epsilon)$ in the estimate of $p$ for any fixed $\mu$ (see, e.g., Eq. 6), and thus also in the estimated boundary, as long as the perturbation is not large enough to "flip" the order of the scores. Eq. 12 similarly implies an $O(\epsilon)$ error in the expected loss.
>
> Additional experiments:
>
> **(a)** For the same **synthetic example** as in Fig. 2, we perturbed the original scores by adding increasing amounts of random noise and then computed the optimal selection rules using Alg. 1 (minimizing 0-1 loss). As expected, these rules achieve perfect predictive parity on the perturbed scores. When evaluated on the original, perfectly calibrated scores, the average fairness violations (differences in PPV and FOR between the groups) do not exceed a few percentage points and indeed follow an approximately linear function in the noise level.
>
> **(b)** We trained a logistic regression model on the **Adult dataset** with “sex” as the protected attribute and calibrated its scores (by group) with isotonic regression. By varying the size of the calibration subset, we induced varying levels of calibration error. We evaluated the resulting predictor on a held-out test set and used our algorithm to compute the optimal classification. On average over 100 repetitions, even if only half of the training set is used to estimate the calibrated scores, the average difference in PPV between the groups is below 3%, and is even smaller for FOR. Again, the violation follows an approximately linear relationship with calibration error.
>
> Figures of both experiments are in https://drive.google.com/file/d/1ixDjtNBDej-AlI3B21N6aJQgxcdFY7xp/view
>
> With respect to the questions and limitations:
>
> **Q1** Yes, the loss-optimal sufficient classifier can lie on a degenerate edge even when there is nontrivial intersection between the feasible sets (see, e.g., the Adult dataset experiment in https://drive.google.com/file/d/1nDHkR3VHPIhAKpFEp0fqWu2_WDnzWAGH/view). Such a point corresponds to a constant classifier on one of the groups. While technically satisfying sufficiency, such cases highlight a limitation of the criterion (see Appendix D).
>
> **Q2/L1** See answer 2 above.
>
> **Q3/L2** See answer 1(c) above.

---

> > ### Author Rebuttal · Reviewer_3gdC · 2026-04-04
> >
> > Thanks for the response, I will keep the positive score.

---

### Official Review · Reviewer_4438 · 2026-03-11

**Soundness:** 3
**Presentation:** 2
**Significance:** 3
**Originality:** 2
**Overall Recommendation:** 4
**Confidence:** 3

**Summary:**

The authors derive a post-processing method for binary classifiers that achieves the fairness constraints of PPV and FOR while minimizing a loss function. To define their approach, they provide a geometric characterization of the set of feasible classifiers satisfying these fairness constraints, which allows them to identify the optimal classifier within this region that optimizes a user-specified objective function or minimizes the deviation from the separation criterion. To achieve sufficiency, their method requires group-wise calibrated scores as well as the sensitive attribute values of the individuals. They apply their approach to the COMPAS dataset.

**Compliance With Llm Reviewing Policy:**

Affirmed.

**Final Justification:**

The authors have addressed my concern, so I will upgrade my score. However, I remain somewhat uncertain, as I did not have time to check all the mathematical proofs.

**Key Questions For Authors:**

1) Could the authors write a conclusion mentioning limitations of their work as well as possible extensions?
2) The theoretical results assume that the scores take finitely many values, whereas in practice classifiers can produce continuous scores. Could the authors clarify the impact of the assumption that only models producing discrete scores (such as decision trees) can be used?
3) The methodology also assumes the existence of an intersection between $\mathcal{C}_0$ and $\mathcal{C}_1$. Could the authors comment on how restrictive this assumption is in practice? Is the intersection always non-empty? In particular, what happens if this intersection does not exist: does the method fail, or is there a possible relaxation or approximation using approximate fairness for example?
4) Would it be possible to compare the results of the proposed approach with at least one existing post-processing method that achieves Sufficiency in Section 6?

**Limitations:**

There are no potential negative societal impacts of their work, and an Impact Statement has been provided. However, the authors could discuss the limitations of their work by adding a Conclusion section.

**Strengths And Weaknesses:**

1) Soundness

Due to the number of review assignments and time constraints, I did not have time to check the appendix and therefore to verify the correctness of the theoretical results.

- For rigor, the authors could add references supporting the fact that scores satisfying the Independence and Separation criteria necessarily lead to decisions that also satisfy these fairness criteria.
- To ease the understanding of the mathematical results on the feasible set of classifiers in the unconstrained learning section, the authors could add the definition of a star-convex set, as well as additional figures illustrating the geometric characterization.
- The theoretical results rely on two important assumptions. The first is that the scores take finitely many values, whereas in practice we usually deal with continuous scores in the [0,1] range. It therefore seems that the method may only be directly applicable to certain algorithms such as decision trees. The second assumption is the existence of an intersection between $\mathcal{C}_0$ and $\mathcal{C}_1$, which could limit the applicability of the methodology.
- Moreover, the algorithm requires group-wise calibrated scores. The authors could therefore add a paragraph in Section 5 providing references on how to obtain multicalibrated scores, as this property is not always guaranteed with unconstrained learning.

2) Presentation

Overall, I find the article well written, with the motivation and contributions clearly stated. Moreover, the paper is well structured: the authors first characterize the feasible region for unconstrained learning and then extend it within the fairness framework. However, a few drawbacks remain:
- There is no conclusion, so the authors do not discuss the limitations or potential extensions of their work.
- One mathematical object, $\phi_{k,l}$, is used in the main text but defined only in the appendix, which makes some parts of Section 4 difficult to follow.

3) Significance

The practical utility and potential impact of the approach are important and clearly highlighted in the paper. The contribution is significant, as fairness post-processing approaches are commonly developed for Independence and Separation criteria, but much less for Sufficiency due to the non-linearity of this constraint.

4) Originality

I find this paper very promising, as the authors derive a geometric characterization of binary classifiers satisfying Sufficiency, which allows practitioners to apply the method regardless of their objective function, including minimizing deviation from the Separation criterion. Moreover, the approach does not require labeled data to be applied, which facilitates its use in practice.
However, some existing post-processing methods are not mentioned in the introduction: [1], [2], [3]. In addition, the authors do not compare their fairness-accuracy trade-off in the experiments with existing approaches that also satisfy the Sufficiency criterion, which makes it difficult to position their method within the literature in terms of fairness metric (PPV and FOR).

[1] Celis, L. E., Huang, L., Keswani, V., & Vishnoi, N. K. (2019). Classification with fairness constraints: A meta-algorithm with provable guarantees. In Proceedings of the 2019 Conference on Fairness, Accountability, and Transparency (FAT '19)* (pp. 319–328). Association for Computing Machinery. https://doi.org/10.1145/3287560.3287586

[2] Zeng, X., Dobriban, E., & Cheng, G. (2022). Fair Bayes-optimal classifiers under predictive parity. In S. Koyejo, S. Mohamed, A. Agarwal, D. Belgrave, K. Cho, & A. Oh (Eds.), Advances in Neural Information Processing Systems (Vol. 35, pp. 27692–27705). Curran Associates, Inc. https://proceedings.neurips.cc/paper_files/paper/2022/file/b1d9c7e7bd265d81aae8d74a7a6bd7f1-Paper-Conference.pdf

[3] Delaney, E., Fu, Z., Wachter, S., Mittelstadt, B., & Russell, C. (2024). OxonFair: A flexible toolkit for algorithmic fairness. In Proceedings of the 38th International Conference on Neural Information Processing Systems (NeurIPS ’24). Curran Associates, Inc.

---

> ### Author Rebuttal · Authors · 2026-03-31
>
> Thank you for your feedback!
>
> Regarding the mentioned points:
>
> **1** Soundness:
>
> * Thank you, we will add the references: the proof that separation is maintained under postprocessing appears in Benger & Ligett (2025), Thm. 17. A similar argument holds for independence.
>
> * We will add the definition of star-convexity.
>
> * Regarding our assumptions, these are two separate points:
>
>   **(a)** Our setting is one where the method is applied on a finite set of score values. Even if the underlying scoring function is continuous, its evaluation on a finite sample yields a finite set of values. Moreover, standard calibration procedures often discretize scores into finitely many bins, and many fairness-related scores are inherently discrete and finite, such as credit scores and admission test scores. We view our ability to handle discrete scores as a significant improvement relative to previous work.
>
>   **(b)** If there is no intersection between the feasible sets, the only possible binary classifier satisfying sufficiency must be degenerate (see Appendix D). In other words, this is not an assumption of the method: a fair binary classifier exists only if there is an intersection.
>
> * Our work assumes that the scores are already group-calibrated (as is the case with COMPAS, for example), not necessarily multicalibrated. Either way, the question we address is not how to obtain fair scores, but how to convert such scores into a fair **binary** decision. If the given scores are not calibrated, we can first calibrate them (e.g., using isotonic regression, like in the experiment below) and then proceed.
>
> **2** Presentation:
> * We will add a conclusion section summarizing: (i) our geometric characterization of binary classifiers satisfying sufficiency and the optimization algorithm; (ii) the practical setting of finite, group-calibrated scores without access to raw data; (iii) robustness to approximate calibration; and (iv) limitations in multi-group settings, where feasibility depends on the intersection of group-specific regions, and directions toward approximate sufficiency.
>
> * $\Phi_{k,l}$ is explained in line 309. It is the quadratic function used to determine which group defines the active boundary on each interval $J_{k,l}$, obtained by comparing the corresponding boundary functions. The coefficient expressions were deferred to Appendix F, as they are cumbersome and not required for the arguments in the main text.
>
> **4** Originality:
>
> Thank you! Our work is most closely related to approaches such as Canetti et al. (2019) and Baumann et al. (2022), which also study exact sufficiency constraints and characterize the structure of optimal classifiers under these constraints. Our contribution complements this line of work by providing an exact geometric characterization and optimal algorithm in the finite-score setting.
>
> As for the mentioned references:
>
> Below we provide a comparison with the OxonFair toolkit [3], using their "cond_use_accuracy" objective (average PPV and FOR difference), on the Adult dataset with "sex" as the protected attribute.
>
> Note that this comparison is possible only in settings where the raw data is available, and in particular where one has access to a labeled training or validation set. Such access is required by [1] and [3], but not by our method. More importantly, these methods address a different problem from ours: they optimize approximate relaxations of the fairness constraints and rely on a tolerance parameter to trade fairness against accuracy. In contrast, our method enforces exact sufficiency, that is, equality of both PPV and FOR, and maximizes accuracy under this strict constraint. Therefore, it yields a single precise operating point.
>
> Despite these differences, the results (summarized below) show that our single operating point achieves comparable (or smaller) fairness violations, while attaining higher accuracy when the OxonFair method operates at similar fairness levels.
>
> | Fairness tol. | Fairness obj. (OxonFair) | Acc. (OxonFair) | Fairness obj. (ours) | Acc. (ours) |
> |---------------|--------------------------|-----------------|----------------------|-------------|
> | 0.010 | **0.0186** | 0.7551 | 0.0218 | **0.7971** |
> | 0.015 | 0.0241 | 0.7737 | **0.0218** | **0.7971** |
> | 0.020 | 0.0295 | 0.7860 | **0.0218** | **0.7971** |
> | 0.025 | 0.0357 | 0.7981 | **0.0218** | 0.7971 |
> | 0.030 | 0.0427 | 0.8077 | **0.0218** | 0.7971 |
> | 0.035 | 0.0483 | 0.8159 | **0.0218** | 0.7971 |
> | 0.040 | 0.0544 | 0.8222 | **0.0218** | 0.7971 |
>
>
> Regarding the work in [2], it considers only equality of PPV, without requiring equality of FOR. This reduces the problem to a one-dimensional condition, whereas our formulation requires matching both PPV and FOR simultaneously, leading to an essentially different (two-dimensional) feasible set and optimization problem. For this reason, the methods are not directly comparable. We would be glad to incorporate references to all three papers in the final version.

---

> > ### Author Rebuttal · Reviewer_4438 · 2026-04-02
> >
> > 4. Originality
> >
> > Q1: In the table, how is the fairness objective calculated? Could the authors clarify what "cond_use_accuracy" refers to and explain why they did not use the "predictive_parity" constraint from OxonFair?
> >
> > Q2: Is it not possible, with your method, to enforce only one of the fairness constraints (PPV or FOR)? Moreover, it should still be possible to compare your method with [2] using the PPV fairness objective, since your method satisfies this constraint.

---

> > > ### Author Response · Authors · 2026-04-06
> > >
> > > In response to your follow-up questions:
> > >
> > > **Q1.** The "cond_use_accuracy" objective in OxonFair measures the average of the absolute differences in PPV and NPV across groups (see https://github.com/oxfordinternetinstitute/oxonfair/blob/master/measures.md; the term "conditional use accuracy" follows Verma & Rubin (2018) https://doi.org/10.1145/3194770.3194776). Since $FOR=1−NPV$, this is equivalent to $\frac{1}{2}(|PPV^0-PPV^1| + |FOR^0-FOR^1|).$
> > >
> > > In contrast, their "predictive_parity" objective measures only $∣PPV^0−PPV^1∣$.
> > >
> > > Since our method enforces equality of **both** PPV and FOR, "cond_use_accuracy" is the most appropriate metric for comparison.
> > >
> > > The "Fairness obj." columns in the table therefore report the value of this quantity for each method.
> > >
> > > **Q2.** Our method is designed to enforce **simultaneous equality of PPV and FOR**, which leads to a genuinely **two-dimensional** problem and requires explicitly characterizing the intersection of the group-wise feasible regions. Algorithm 1 is built to trace this two-dimensional boundary and optimize objectives over it.
> > >
> > > If one enforces only a single constraint (e.g., PPV equality), the problem reduces to a simpler, one-dimensional condition, and the geometric structure we develop (e.g., identifying the active boundary across segments) is no longer needed. In that sense, applying our method to this setting is possible but unnecessarily complex. This would require a specially structured objective $\mathcal F$ in Algorithm 1, which effectively collapses the problem onto a single axis, thereby discarding the two-dimensional structure that the algorithm is designed to trace.
> > >
> > > Moreover, the structure of the optimal solutions is fundamentally different in the two settings. In our work, we show that enforcing both PPV and FOR equality may require a selection rule that is a threshold for one group but a non-threshold (possibly randomized) rule for the other. In contrast, when enforcing only PPV equality, it suffices to consider group-specific threshold rules, and the solution can be obtained via a one-dimensional search exploiting the monotonicity of PPV under thresholding (as in [2]).
> > >
> > > Nevertheless, for completeness, we performed such a comparison on the Adult dataset. As expected, our original method achieves slightly lower accuracy than [2], due to the additional constraint on FOR; however, when enforcing only PPV equality, our method achieves similar PPV difference and accuracy to [2], indicating that the difference between the methods stems from the additional FOR constraint:
> > >
> > > | Method | Accuracy | $\|PPV^0 - PPV^1\|$ | $\|FOR^0 - FOR^1\|$ |
> > > |--------|----------|---------------------|---------------------|
> > > | Ours (PPV & FOR) | 0.7971 | 0.0378 | 0.0057 |
> > > | Ours (only PPV) | 0.8223 | 0.0385 | 0.1048 |
> > > | FairBayes-DPP [2] | 0.8299 | 0.0401 | 0.1080 |

---

### Official Review · Reviewer_J4an · 2026-03-12

**Soundness:** 3
**Presentation:** 3
**Significance:** 3
**Originality:** 4
**Overall Recommendation:** 4
**Confidence:** 2

**Summary:**

This paper studies the problem of optimal binary classification under the sufficiency (predictive parity) constraint with discrete group-calibrated scores. The core contribution lies in transforming the problem into finding the intersection of feasible regions for two groups in a two-dimensional plane (with PPV and FOR as coordinates). The authors provide a piecewise analytical expression for the boundary of this intersection and, based on this, propose Algorithm 1, which can efficiently solve two types of optimization problems under the sufficiency constraint: loss minimization and deviation from separation minimization. Compared to the work of Baumann et al. (2022), the main advancement of this paper is the removal of the assumption of continuous score distribution, making the method applicable to discrete scoring systems like COMPAS.

**Compliance With Llm Reviewing Policy:**

Affirmed.

**Key Questions For Authors:**

(1) When using an estimator in place of the true distribution $P(Y=1 | S, A)$, if the estimation error is $\epsilon$, does a theoretical bound exist for the degree of violation of the group calibration constraint and the loss in overall utility for the final decision rule? Please supplement the corresponding theoretical analysis or numerical experiments to evaluate the method's sensitivity to estimation error.

(2) Regarding the parameters $l_{01}, l_{10}$, please supplement sensitivity analysis experiments or provide theoretical justification for what constitutes an optimal or reasonable choice.

(3) For cases involving multi-class or continuous variables, how do the algorithm's computational complexity and the likelihood of an empty feasible region intersection change as the number of groups increases? Please supplement analysis or experiments to clarify the scalability bottlenecks of this method in real-world multi-group scenarios.

(4) Please conduct experiments using more realistic and common data. For instance, when $S$ is a model score trained on finite data, how does the method handle this, and does the resulting decision rule still satisfactorily satisfy sufficiency?

**Limitations:**

The paper lacks a concluding chapter that summarizes the work and discusses its limitations in depth. It is recommended that the authors supplement this content.

**Strengths And Weaknesses:**

Strengths

Transforming an abstract fairness-constrained problem into an intuitive geometric framework. One of the most prominent contributions is the reformulation of the optimal classification problem under sufficiency constraints into a geometric problem on a two-dimensional plane. Using PPV and FOR as axes, the authors precisely characterize the structure of the single-group feasible region (star-convexity, piecewise hyperbolic boundary) and equate the sufficiency constraint to finding the intersection of feasible regions for two groups. This geometric perspective not only clarifies the problem's structure but also provides a clear framework for subsequent algorithm design—ultimately reducing an abstract fairness optimization problem to elementary operations like solving quadratic equations segment-by-segment on the intersection boundary.

A complete theoretical chain from modeling to algorithm demonstrates high theoretical integrity and internal consistency. The paper presents a complete theoretical chain from problem formulation to algorithm design. This includes the geometric characterization of the single-group feasible region (Theorems 3.3, 3.4), the analytical expression for the multi-group intersection boundary (Section 4), and the solution of two optimization problems: loss minimization (Section 5) and deviation from separation minimization (Section 6). Each step is supported by rigorous mathematical derivation. Notably, two seemingly different optimization objectives are ultimately unified under the same algorithmic framework (Algorithm 1), reflecting the internal consistency of the theoretical system.

Breaking the strong assumption of continuous score distribution from prior work, significantly expanding the method's practical applicability. While Baumann et al. (2022) proposed a post-processing method for sufficiency, their analysis relied on the strong assumption of a continuous score distribution with full support, preventing its application to real-world systems like COMPAS that output discrete, finite scores. This paper explicitly addresses the practical scenario of discrete, finite-valued scores, without relying on any continuity assumptions. It fills an important gap between theory and practice, significantly expanding the applicability of sufficiency-based methods.

Weaknesses

(1) In constructing the decision rule, the accuracy of the conditional distribution $P(Y=1 | S, A)$ is crucial. The authors fail to discuss the potential impact on the fairness satisfaction and utility of the decision rule when using estimated values instead of the true distribution.

(2) The construction of the decision rule relies on finding the boundary of the intersection of feasible regions for each group, which is demonstrated for the binary-group case. However, in the real world, protected attributes can be multi-class or continuous, posing significant challenges for this method, which the authors do not discuss in depth.

(3) The data used in the experiments approximately satisfy sufficiency, which greatly reduces the difficulty of achieving group calibration, making it hard to generalize the findings to more common real-world data.

(4) The paper's structure is incomplete, lacking a concluding chapter.

---

> ### Author Rebuttal · Authors · 2026-03-30
>
> Thank you for your feedback!
>
> Regarding the mentioned weaknesses and questions:
>
> **W1/Q1** We note that our assumption of perfect calibration is actually weaker than the one typically made in the literature (e.g., Hardt, Price & Srebro (2016); Canetti et al. (2019); Baumann et al. (2022)), which assumes knowledge of the true probabilities $P(Y=1|X)$. Nonetheless, we agree that it is interesting to explore the impact of using approximately group-calibrated scores. Here is the sketch of a  simple analysis and two additional experiments, versions of which we could incorporate into the final version of the paper. These suggest that the impact of imperfectly group-calibrated scores is expected to be small and approximately linear in the distance from group-calibration.
>
> Assuming the scores $s$ are estimated with error $\epsilon$, our theoretical analysis implies an error of $O(\epsilon)$ in the estimate of $p$ for any fixed $\mu$ (see, e.g., Eq. 6), and thus also in the estimated boundary, as long as the perturbation is not large enough to "flip" the order of the scores. Eq. 12 similarly implies an $O(\epsilon)$ error in the expected loss.
>
> Additional experiments:
>
> **(a)** For the same **synthetic example** as in Fig. 2, we perturbed the original scores by adding increasing amounts of random noise and then computed the optimal selection rules using Alg. 1 (minimizing 0-1 loss). As expected, these rules achieve perfect predictive parity on the perturbed scores. When evaluated on the original, perfectly calibrated scores, the average fairness violations (differences in PPV and FOR between the groups) do not exceed a few percentage points and indeed follow an approximately linear function in the noise level.
>
> **(b)** We trained a logistic regression model on the **Adult dataset** with “sex” as the protected attribute and calibrated its scores (by group) with isotonic regression. By varying the size of the calibration subset, we induced varying levels of calibration error. We evaluated the resulting predictor on a held-out test set and used our algorithm to compute the optimal classification. On average over 100 repetitions, even if only half of the training set is used to estimate the calibrated scores, the average difference in PPV between the groups is below 3%, and is even smaller for FOR. Again, the violation follows an approximately linear relationship with calibration error.
>
> Figures of both experiments are in https://drive.google.com/file/d/1ixDjtNBDej-AlI3B21N6aJQgxcdFY7xp/view
>
> **Q2** Note that $\ell_{01}$ and $\ell_{10}$ are not parameters of the model. Rather, they are the loss parameters. For example, 0-1 loss corresponds to $\ell_{01}=\ell_{10}=1$; minimizing this loss corresponds to maximizing the accuracy. Changing these parameters only "moves" the optimal point along the intersection's boundary.
>
> **W2/Q3** For more than two groups, the extension of our algorithm is straightforward whenever the intersection is nonempty: the boundary is defined by the pointwise maximum of all the group-specific boundary functions (see Eq. 10), and the same piecewise analysis applies. The computational complexity thus scales linearly with the number of groups, up to the number of boundary segments.
>
> Whether the intersection itself exists -- or, equivalently, whether there exists a binary classifier satisfying sufficiency for all groups -- is a structural property of the given score distributions, determined by the relative geometry of the group-specific feasible sets. As the number of groups increases, the intersection involves more sets and is therefore more restrictive, and more likely to be empty.
>
> **W3/Q4** As mentioned in the answer to W1/Q1, we performed an additional experiment using the Adult dataset.
>
> We emphasize that the focus of our work is not on how to calibrate scores, but on how to convert calibrated scores into a fair **binary** decision rule. The fact that the COMPAS scores are already approximately group-calibrated is not substantive: if they have not been, we could first apply a standard calibration procedure, such as isotonic regression.
>
> **W4** We will add a conclusion summarizing: (i) our geometric characterization of binary classifiers satisfying sufficiency and the resulting optimization algorithm; (ii) the practical setting of finite, group-calibrated scores without access to raw data; (iii) robustness to approximate calibration; and (iv) limitations in multi-group settings, where feasibility depends on the intersection of group-specific regions, along with directions toward approximate sufficiency.

---

> > ### Author Rebuttal · Reviewer_J4an · 2026-04-01
> >
> > Thank you for the author's response. I will improve my score.

---

### Decision · Program_Chairs · 2026-04-30

**Decision:**

Accept (regular)

**Comment:**

The reviewers agree the paper studies optimal binary classification under sufficiency when scores are finite-valued and group-calibrated with a sound theoretical framework that is also practical. There are also concerns on the assumptions, analyses, practicality, experiments, related work, and presentation.

During the rebuttal, the authors provided detailed explanations that addressed most of the reviewer concerns. Two reviewers did show low confidence, and one reviewer still questions the overall empirical support and strong assumptions of perfectly calibrated scores while still supporting the paper.

Overall, all the scores are clearly positive, so the recommendation is to accept the paper.